# Improved Balanced Classification with Theoretically Grounded Loss Functions

**Corinna Cortes**
Google Research
New York, NY 10011
corinna@google.com

**Mehryar Mohri**
Google Research & CIMS
New York, NY 10011
mohri@google.com

**Yutao Zhong**
Google Research
New York, NY 10011
yutaozhong@google.com

## Abstract

The *balanced loss* is a widely adopted objective for multi-class classification under class imbalance. By assigning equal importance to all classes, regardless of their frequency, it promotes fairness and ensures that minority classes are not overlooked. However, directly minimizing the balanced classification loss is typically intractable, which makes the design of effective surrogate losses a central question. This paper introduces and studies two advanced surrogate loss families: Generalized Logit-Adjusted (GLA) loss functions and Generalized Class-Aware weighted (GCA) losses. GLA losses generalize Logit-Adjusted losses, which shift logits based on class priors, to the broader general cross-entropy loss family. GCA loss functions extend the standard class-weighted losses, which scale losses inversely by class frequency, by incorporating class-dependent confidence margins and extending them to the general cross-entropy family. We present a comprehensive theoretical analysis of consistency for both loss families. We show that GLA losses are Bayes-consistent, but only $\mathcal{H}$-consistent for complete (i.e., unbounded) hypothesis sets. Moreover, their $\mathcal{H}$-consistency bounds depend inversely on the minimum class probability, scaling at least as $1/p_{\min}$. In contrast, GCA losses are $\mathcal{H}$-consistent for any hypothesis set that is bounded or complete, with $\mathcal{H}$-consistency bounds that scale more favorably as $1/\sqrt{p_{\min}}$, offering significantly stronger theoretical guarantees in imbalanced settings. We report the results of experiments demonstrating that, empirically, both the GCA losses with calibrated class-dependent confidence margins and GLA losses can greatly outperform straightforward class-weighted losses as well as the LA losses. GLA generally performs slightly better in common benchmarks, whereas GCA exhibits a slight edge in highly imbalanced settings. Thus, we advocate for both GLA and GCA losses as principled, theoretically sound, and state-of-the-art surrogates for balanced classification under class imbalance.

## 1 Introduction

Class imbalance is a prevalent challenge in real-world multi-class classification problems. Applications such as medical diagnosis, fraud detection, and rare event prediction often involve highly skewed label distributions, where a small subset of classes dominate the data, while others, sometimes the most critical, are heavily underrepresented. Standard training objectives, such as minimizing the unweighted cross-entropy loss, tend to be biased toward majority classes, leading to poor performance on minority classes and undermining the fairness, soundness and reliability of learned models.

To address this issue, a widely studied approach is to minimize the *balanced loss*, which assigns equal importance to all classes regardless of their frequency in the training data [Chan and Stolfo, 1998, Brodersen et al., 2010, Kotlowski et al., 2011, Menon et al., 2013, Cao et al., 2019, Menon

et al., 2021, Cui et al., 2019]. This promotes fairness by equalizing performance across demographic groups [Khalili et al., 2023, Hardt et al., 2016] and ensures that minority classes are not overlooked in long-tailed datasets [Feldman, 2020, Zhang et al., 2023] (see Appendix A). It is also crucial in federated learning, where data imbalances across clients can lead to biased models that favor heavy users [Li et al., 2021, McMahan et al., 2017, Mohri et al., 2019]. By reweighting the loss contributions from different classes, the balanced loss promotes equitable treatment of all labels and has been shown to better align with metrics such as balanced accuracy and macro-F1. However, directly optimizing the balanced classification loss is typically intractable in practice. Thus, the design of effective surrogate losses that are tractable to optimize is a central challenge in imbalanced learning.

This paper introduces and studies two families of surrogate losses: Generalized Logit-Adjusted (GLA) loss functions and Generalized Class-Aware weighted (GCA) losses. GLA losses generalize Logit-Adjusted losses [Menon et al., 2021], which shift logits based on class priors, to the broader general cross-entropy loss family [Mao et al., 2023f]. GCA loss functions extend the standard class-weighted losses, which scale losses inversely by class frequency, by incorporating class-dependent confidence margins and extending them to the general cross-entropy family.

We present a comprehensive theoretical analysis of their consistency. We show that GLA losses are Bayes-consistent [Zhang, 2004a, Bartlett et al., 2006, Zhang, 2004b, Tewari and Bartlett, 2007, Steinwart, 2007], but only $\mathcal{H}$-consistent [Awasthi et al., 2022a,b, Mao et al., 2023f,b] for complete (i.e., unbounded) hypotheses. Moreover, their $\mathcal{H}$-consistency bounds depend inversely on the minimum class probability, $p_{\min}$, scaling at least as $1/p_{\min}$. In contrast, GCA losses are $\mathcal{H}$-consistent for any hypothesis set that is bounded or complete, with $\mathcal{H}$-consistency bounds that scale more favorably as $1/\sqrt{p_{\min}}$, offering significantly stronger theoretical guarantees in imbalanced settings.

We also report the results of experiments demonstrating that, empirically, both the GCA losses with calibrated class-dependent confidence margins and GLA losses comfortably outperform straightforward class-weighted losses as well as the LA losses. GLA generally performs slightly better in common benchmarks, whereas GCA exhibits a slight edge in highly imbalanced settings.

Taken together, our results establish GLA and GCA losses as theoretically grounded and practically effective classification algorithms for tackling class imbalance in multi-class learning. Their complementary strengths make them well-suited for a wide range of real-world applications where fairness across classes is paramount.

The rest of this paper is structured as follows. Section 3 reviews fundamental concepts related to class imbalance in multi-class classification, introduces the balanced loss (Section 3.1), discusses existing surrogate losses (Section 3.2), and highlights the limitations of current approaches (Section 3.3). Section 4 introduces two novel surrogate loss families: Generalized Logit-Adjusted (GLA) (Section 4.1) and Generalized Class-Aware weighted (GCA) losses (Section 4.2). A comprehensive theoretical analysis of their consistency and margin bounds is provided in Section 5 and Appendix B. Finally, Section 6 reports empirical results on CIFAR-10, CIFAR-100, and Tiny ImageNet, demonstrating the effectiveness of our algorithms, which are based on the minimization of these loss functions.

## 2 Preliminaries

Let $\mathcal{X}$ denote the input space and $\mathcal{Y} = [n] \coloneqq \{1, \ldots, n\}$ represent the set of $n$ possible labels. We consider a data distribution $\mathcal{D}$ over the combined input-label space $\mathcal{X} \times \mathcal{Y}$. Our hypothesis set, denoted by $\mathcal{H}$, consists of functions that map an input-label pair $(x, y)$ to a real-valued score, $h: \mathcal{X} \times \mathcal{Y} \to \mathbb{R}$. We denote by $p(x)$ the marginal probability density of an input $x$, and by $p(y)$ the marginal probability of a class label $y$. The minimum class marginal is defined as $p_{\min} = \min_{y \in \mathcal{Y}} p(y)$. The conditional distributions $p(x \mid y)$ and $p(y \mid x)$ represent the probability of input $x$ given label $y$, and label $y$ given input $x$, respectively.

Let $\mathcal{H}_{\mathrm{all}}$ denote the set of all measurable functions, and a $\ell: \mathcal{H}_{\mathrm{all}} \times \mathcal{X} \times \mathcal{Y} \to \mathbb{R}$ the loss function adopted to penalize inaccurate predictions. Then, the *generalization error* of a hypothesis $h \in \mathcal{H}$ is defined as its expected loss: $\mathcal{R}_\ell(h) = \mathbb{E}_{(x,y)\sim\mathcal{D}}[\ell(h, x, y)]$. The lowest possible generalization error achievable within the hypothesis set $\mathcal{H}$ is the *best-in-class generalization error*, $\mathcal{R}_\ell^*(\mathcal{H}) = \inf_{h\in\mathcal{H}} \mathcal{R}_\ell(h)$.

For any input $x \in \mathcal{X}$, a hypothesis $h \in \mathcal{H}$ assigns a predicted label $h(x)$ by selecting the class with the highest score: $h(x) = \operatorname{argmax}_{y\in\mathcal{Y}} h(x, y)$ (ties are broken by choosing the highest index). The

standard *zero-one loss function* for multi-class classification is defined as $\ell_{0-1}(h, x, y) \coloneqq \mathbb{1}_{h(x) \neq y}$, which is 1 if the prediction is incorrect and 0 otherwise.

The *margin* $\rho_h(x, y)$ for a predictor $h \in \mathcal{H}$ on a labeled example $(x, y)$ measures the confidence of the correct prediction: $\rho_h(x, y) = h(x, y) - \max_{y' \neq y} h(x, y')$. This is the difference between the score of the true label $y$ and the highest score among all other labels $y'$.

The generalization error of a hypothesis $h$ can also be expressed as the expectation of the *conditional error* over the input $x$: $\mathcal{R}_\ell(h) = \mathbb{E}_x[\mathcal{C}_\ell(h, x)]$, where $\mathcal{C}_\ell(h, x) = \sum_{y \in \mathcal{Y}} \mathsf{p}(y \mid x) \ell(h, x, y)$. The *best-in-class conditional error* is $\mathcal{C}_\ell^*(\mathcal{H}, x) = \inf_{h \in \mathcal{H}} \mathcal{C}_\ell(h, x)$. The difference, $\Delta\mathcal{C}_{\ell, \mathcal{H}}(h, x) = \mathcal{C}_\ell(h, x) - \mathcal{C}_\ell^*(\mathcal{H}, x)$, is termed the *conditional regret* for the loss function $\ell$. These concepts and definitions are useful in our analysis of the consistency of loss functions.

# 3 Background and Related Work

We first review fundamental concepts related to class imbalance in multi-class classification, introduce the balanced loss, discuss existing surrogate losses, and highlight the limitations of current approaches.

## 3.1 Class Imbalance and Balanced Loss

Class imbalance in multi-class settings arises when the label distribution $\mathsf{p}(y)$ is highly skewed, with some classes (often referred to as "tail" labels) having much lower probabilities of occurrence compared to others (the "head" or majority classes). In such cases, many recent studies [Chan and Stolfo, 1998, Brodersen et al., 2010, Kotlowski et al., 2011, Menon et al., 2013, Cao et al., 2019, Menon et al., 2021, Cui et al., 2019] suggest that the balanced loss ($\ell_{\mathrm{BAL}}$) is a more appropriate loss function than the standard zero-one loss. The balanced loss assigns equal importance to all classes, irrespective of their frequency, and is thus viewed as promoting fairness by equalizing performance across demographic groups [Khalili et al., 2023, Hardt et al., 2016, Conitzer et al., 2019] and ensuring minority classes are not overlooked in long-tailed datasets [Feldman, 2020, Zhang et al., 2023] (see Appendix A). It is also crucial in federated learning, where data imbalances across clients can lead to biased models that favor majority users [Li et al., 2021, McMahan et al., 2017, Mohri et al., 2019].

The balanced loss reduces the influence of class imbalances by averaging the per-class loss by weighting the error for each example $(h, x, y)$ by the inverse of the probability of the true class $\mathsf{p}(y)$:

$$\ell_{\mathrm{BAL}}(h, x, y) = \frac{\mathbb{1}_{h(x) \neq y}}{\mathsf{p}(y)}. \tag{1}$$

The following lemma characterizes the best-in-class conditional error and the corresponding conditional regret for the balanced loss. For any input $x \in \mathcal{X}$, we denote by $\mathsf{H}(x)$ the set of labels that can be predicted by hypotheses in $\mathcal{H}$ for that input: $\mathsf{H}(x) = \{h(x) : h \in \mathcal{H}\}$. The proof of Lemma 1 is provided in Appendix D.

**Lemma 1.** *For any $x \in \mathcal{X}$, the best-in-class conditional error and the conditional regret for $\ell_{\mathrm{BAL}}$ can be expressed as follows:*

$$\mathcal{C}_{\ell_{\mathrm{BAL}}}^*(\mathcal{H}, x) = \sum_{y \in \mathcal{Y}} \frac{\mathsf{p}(y \mid x)}{\mathsf{p}(y)} - \max_{y \in \mathsf{H}(x)} \frac{\mathsf{p}(y \mid x)}{\mathsf{p}(y)} \qquad \Delta\mathcal{C}_{\ell_{\mathrm{BAL}}, \mathcal{H}}(h, x) = \max_{y \in \mathsf{H}(x)} \frac{\mathsf{p}(y \mid x)}{\mathsf{p}(y)} - \frac{\mathsf{p}(h(x)) \mid x)}{\mathsf{p}(h(x))}.$$

## 3.2 Existing Surrogate Losses for Balanced Learning

Several surrogate losses have been proposed for optimizing the balanced loss. Here, we review two prominent Bayes-consistent examples:

**Class-Weighted Cross-Entropy**: A common strategy is to use the class-weighted cross-entropy loss [Xie and Manski, 1989, Morik et al., 1999], which adjusts the standard cross-entropy loss by weighting each example inversely proportional to its class frequency $\mathsf{p}(y)$:

$$\ell_{\mathrm{WCE}}(h, x, y) = -\frac{1}{\mathsf{p}(y)} \log\left(\frac{e^{h(x, y)}}{\sum_{y' \in \mathcal{Y}} e^{h(x, y')}}\right). \tag{2}$$

As pointed by [Byrd and Lipton, 2019], the limitation of $\ell_{\mathrm{WCE}}$ is that in separable cases, class-weighted cross-entropy may still yield solutions with zero training loss that do not adjust decision

boundaries meaningfully toward minority or majority classes. This is because class weighting does not influence the classifier once perfect separation is achieved. As a result, the method fails to address imbalance in such regimes.

**Logit-Adjusted (LA) Losses**: More recently, Menon et al. [2021] introduced Logit-Adjusted (LA) losses. These losses modify the logits (outputs before softmax) based on class priors, typically by adding a term $\tau \log(\mathsf{p}(y))$ with $\tau > 0$:

$$\ell_{\mathrm{LA}}(h, x, y) = -\log\left(\frac{e^{h(x,y)+\tau \log(\mathsf{p}(y))}}{\sum_{y'\in\mathcal{Y}} e^{h(x,y')+\tau \log(\mathsf{p}(y'))}}\right). \tag{3}$$

As we will show in Section 5, $\ell_{\mathrm{LA}}$ is not Bayes-consistent for the balanced loss when $\tau \neq 1$.

A detailed discussion of other approaches for handling class imbalance, including alternative loss weighting schemes [Cui et al., 2019, Fan et al., 2017, Jamal et al., 2020, Wang et al., 2023, 2025, Li et al., 2025], margin modifications [Masnadi-Shirazi and Vasconcelos, 2010, Iranmehr et al., 2019, Zhang et al., 2017, Cao et al., 2019, Tan et al., 2020, Jiawei et al., 2020], data augmentation and sampling techniques [Kubat and Matwin, 1997, Wallace et al., 2011, Chawla et al., 2002, Yin et al., 2018], threshold adjustments [Fawcett and Provost, 1996, Provost, 2000, Maloof, 2003, King and Zeng, 2001, Collell et al., 2016, Menon et al., 2021, Zhu et al., 2023], and weight normalization methods [Zhang et al., 2019a, Kim and Kim, 2019, Kang et al., 2020] is included in Appendix A.

### 3.3 Limitations of Existing Approaches

Despite their usefulness, existing surrogate losses and related methods admit some limitations. *Class-weighted cross-entropy* often has a minimal effect in settings where data is easily separable. In such cases, solutions that achieve zero training loss (perfect separation) remain optimal even with class weighting, failing to shift decision boundaries effectively towards dominant classes as might be desired [Byrd and Lipton, 2019]. *Logit-Adjusted (LA) losses*, as we will demonstrate in Section 5, are not Bayes-consistent for the balanced loss when the temperature parameter $\tau \neq 1$. Consequently, optimal tuning of $\tau$ often lacks a theoretical guarantee, and the method itself offers limited flexibility. *Other margin modification techniques* [e.g., Cao et al., 2019, Tan et al., 2020] may not be Bayes-consistent for the balanced loss, even in simpler binary classification problems [Menon et al., 2021]. *The drawbacks of other strategies* beyond direct loss modification, such as weight normalization, have also been previously noted [Menon et al., 2021].

## 4 Surrogate Loss Families

This section generalizes two surrogate loss families designed for learning with class imbalance: Generalized Logit-Adjusted (GLA) loss functions and Generalized Class-Aware weighted (GCA) losses. Both families are derived from the general cross-entropy (GCE) framework [Mao et al., 2023f]. For any $(h, x, y) \in \mathcal{H} \times \mathcal{X} \times \mathcal{Y}$, the GCE loss is defined as:

$$\ell_{\mathrm{GCE}}(h, x, y) = \Psi^q\left(\frac{e^{h(x,y)}}{\sum_{y'\in\mathcal{Y}} e^{h(x,y')}}\right), \quad \text{with} \quad \Psi^q(t) = \begin{cases} -\log(t) & \text{if } q = 0 \\ \frac{1}{q}(1 - t^q) & \text{if } q \in (0, \infty). \end{cases}$$

Specific choices of $q$ recover well-known loss functions: $q = 0$ yields the *logistic loss* (or standard cross-entropy) [Verhulst, 1838, 1845, Berkson, 1944, 1951]; $q \in (0, 1)$ gives the *generalized cross-entropy loss* notable for its robustness to label noise [Zhang and Sabuncu, 2018]; and $q = 1$ corresponds to the *mean absolute error loss* [Ghosh et al., 2017].

### 4.1 Generalized Logit-Adjusted (GLA) Losses

A *Generalized Logit-Adjusted (GLA) Loss* modifies the logits within the GCE family by incorporating a class-prior-based bias term, $\log(\mathsf{p}(y))/(1 - q)$:

$$\ell_{\mathrm{GLA}}(h, x, y) = \Psi^q\left(\frac{e^{h(x,y)+\frac{\log(\mathsf{p}(y))}{1-q}}}{\sum_{y'\in\mathcal{Y}} e^{h(x,y')+\frac{\log(\mathsf{p}(y'))}{1-q}}}\right), \tag{4}$$

The GLA loss family generalizes the Logit-Adjusted (LA) loss with $\tau = 1$. Specifically, when $q = 0$, Eq. (4) recovers the LA loss with $\tau = 1$ previously defined in Eq. (3). Thus, GLA extends the concept

of logit adjustment to the broader GCE family. As will be detailed in Section 5.2, GLA losses are Bayes-consistent for any $q \in [0, 1)$, offering greater flexibility compared to the original LA loss (whose limitations were discussed in Section 3.3).

The term inside the $\Psi^q$ function in Eq. (4) can be rewritten to highlight its behavior:

$$\frac{e^{h(x,y)+\frac{\log(\mathsf{p}(y))}{1-q}}}{\sum_{y' \in \mathcal{Y}} e^{h(x,y')+\frac{\log(\mathsf{p}(y'))}{1-q}}} = \frac{e^{h(x,y)} \cdot \mathsf{p}(y)^{\frac{1}{1-q}}}{\sum_{y' \in \mathcal{Y}} e^{h(x,y')} \cdot \mathsf{p}(y')^{\frac{1}{1-q}}} = \frac{1}{\sum_{y' \in \mathcal{Y}} e^{h(x,y')-h(x,y)} \cdot \left(\frac{\mathsf{p}(y')}{\mathsf{p}(y)}\right)^{\frac{1}{1-q}}}.$$

In this formulation, the term $\left(\mathsf{p}(y')/\mathsf{p}(y)\right)^{\frac{1}{1-q}}$ acts as a weighting factor in the denominator, effectively creating a pairwise label margin adjustment that depends on the relative frequencies of class $y$ (the true class) and other classes $y'$. This mechanism encourages a larger separation (margin) when $y$ is a rare class (low $\mathsf{p}(y)$) and $y'$ is a dominant class (high $\mathsf{p}(y')$) and reduces the risk that scores for dominant classes overshadow those for rare classes.

## 4.2 Generalized Class-Aware (GCA) Losses

A *Generalized Class-Aware (GCA) loss* introduces class sensitivity by inversely weighting the GCE loss by class frequency $\mathsf{p}(y)$ and incorporating class-dependent confidence margins $\rho_y$:

$$\ell_{\mathrm{GCA}}(h, x, y) = \frac{1}{\mathsf{p}(y)} \Psi^q \left( \frac{e^{h(x,y)/\rho_y}}{\sum_{y' \in \mathcal{Y}} e^{h(x,y')/\rho_y}} \right), \tag{5}$$

where $\boldsymbol{\rho} = (\rho_1, \ldots, \rho_n)$ is a vector of positive confidence margin parameters for each class. The GCA formulation encompasses standard class-weighting as a special case. For instance, the class-weighted cross-entropy loss (Eq. (2)) is recovered when $q = 0$ and all confidence margins $\rho_y$ are set to 1. If all $\rho_y = 1$, Eq. (5) simplifies to: $\ell_{\mathrm{GCA}}(h, x, y) = \frac{1}{\mathsf{p}(y)} \Psi^q \left( \frac{e^{h(x,y)}}{\sum_{y' \in \mathcal{Y}} e^{h(x,y')}} \right)$, thereby extending the class-weighted cross-entropy concept to the entire GCE family. The motivation for using the inverse of the prior in GCA remains the same for $q \neq 1$ as for $q = 1$. The parameter $q$ simply specifies a particular loss within the generalized cross-entropy family, applicable in both standard and imbalanced settings. The inverse of the prior is used to align with the definition of the balanced loss, which reduces the influence of class imbalance by reweighting each example's error accordingly. This ensures that GCA losses benefit from consistency guarantees with respect to the balanced loss.

The introduction of distinct confidence margin parameters $\boldsymbol{\rho}$ is a key aspect of GCA losses. These parameters allow for fine-tuned adjustments to the decision boundaries. By applying class-specific scaling with factors related to $\rho_y$ to the logit differences $[h(x, y) - h(x, y')]$-terms that inherently represent margins, the GCA loss (through an effective transformation to $(h(x, y) - h(x, y'))/\rho_y$) can more effectively separate dominant and rare classes, as such transformation modulates how confidently each class needs to be separated. Such margin adjustments, as highlighted by recent work of Cortes et al. [2025], play a crucial role in effectively shifting decision boundaries across classes and mitigating imbalance. This, in turn, addresses the limitations of simpler class-weighting schemes mentioned in Section 3.3.

Note that while the $\rho_k$ values can be treated as tunable hyperparameters and freely tuned via cross-validation, the search can be effectively guided by focusing on vectors $[\rho_k]_k$ near $[m_k^{1/3}]_k$, where $m_k$ denotes the number of samples in class $k$, as suggested by Cortes et al. [2025] and followed in our experiments. A similar derivation to theirs, adaptable to our setting, shows these values are theoretically optimal in a separable case, providing justification and guidance for selecting $\rho_k$ for GCA losses. Empirically, we also found GCA losses to be robust to variations in $\rho_k$ around these values. Consequently, while $\rho_k$ can be tuned, the default choice of $m_k^{1/3}$ performs well. When the number of classes $n$ is large, the search space can be further reduced by assigning identical $\rho_k$ values to underrepresented classes and reserving distinct values for the most frequent ones.

For fixed hyperparameters, the computational cost of GLA and GCA losses is comparable to that of standard neural networks trained with cross-entropy loss (that is, logistic loss with softmax) and to that of the baselines. Our loss functions are adapted from the general cross-entropy family and both share similar convergence behavior and remain practical when optimized with commonly used optimizers such as SGD, Adam, and AdaGrad. While our methods introduce additional hyperparameters, namely

$\rho_k$ and $q$ in GCA losses and $q$ in GLA losses, the value of $\rho_k$ has a default choice (as discussed above), and $q$ serves a similar role to hyperparameters in the baseline methods listed in Table 1 in Section 6, many of which also involve at least one extra tunable parameter.

# 5 Theoretical Analysis

In this section, we leverage Lemma 1 to present a comprehensive theoretical analysis of the consistency for the two proposed surrogate loss families: Generalized Logit-Adjusted (GLA) losses and Generalized Class-Aware (GCA) losses.

## 5.1 Consistency Notions

A critical characteristic of a surrogate loss function $\ell_A$, used in place of a target loss function $\ell_B$, is its *Bayes-consistency* [Steinwart, 2007]. This property ensures that if a sequence of predictor $\{h_n\}_{n\in\mathbb{N}}$ within $\mathcal{H}_{\mathrm{all}}$ (the set of all measurable functions) asymptotically minimizes the surrogate loss $\ell_A$, it will also asymptotically minimize the target loss $\ell_B$. Formally: $\lim_{n\to+\infty}\mathcal{R}_{\ell_A}(h_n) = \mathcal{R}_{\ell_A}^*(\mathcal{H}_{\mathrm{all}}) \Rightarrow \lim_{n\to+\infty}\mathcal{R}_{\ell_B}(h_n) = \mathcal{R}_{\ell_B}^*(\mathcal{H}_{\mathrm{all}})$. However, Bayes-consistency is an asymptotic concept and is defined only for the comprehensive class of all measurable functions $\mathcal{H}_{\mathrm{all}}$. A more practically relevant and informative concept is that of $\mathcal{H}$*-consistency bounds*. These bounds are non-asymptotic and tailored to a specific hypothesis class $\mathcal{H}$ [Awasthi et al., 2022a,b, 2021a,b, 2023a,b, Mao et al., 2023a,b,c,d,e,f, 2024a,b,c,d,e,f,g, Mohri et al., 2024, Cortes et al., 2024, Mao et al., 2025c,a,b, Mao, 2025, Zhong, 2025, DeSalvo et al., 2025]). In the realizable setting, these bounds take the form:

$$\forall h \in \mathcal{H}, \quad \mathcal{R}_{\ell_B}(h) - \mathcal{R}_{\ell_B}^*(\mathcal{H}) \leq \Gamma\big(\mathcal{R}_{\ell_A}(h) - \mathcal{R}_{\ell_A}^*(\mathcal{H})\big).$$

Here, $\Gamma$ is a non-increasing concave function such that $\Gamma(0) = 0$. In the more general non-realizable setting, the bound is augmented by a *minimizability gap*, $\mathcal{M}_\ell(\mathcal{H}) = \mathcal{R}_\ell^*(\mathcal{H}) - \mathbb{E}_x[\mathcal{C}_\ell^*(\mathcal{H}, x)]$. This gap quantifies the difference between the best-in-class error and the expected best-in-class conditional error. The augmented bound is:

$$\mathcal{R}_{\ell_B}(h) - \mathcal{R}_{\ell_B}^*(\mathcal{H}) + \mathcal{M}_{\ell_B}(\mathcal{H}) \leq \Gamma\big(\mathcal{R}_{\ell_A}(h) - \mathcal{R}_{\ell_A}^*(\mathcal{H}) + \mathcal{M}_{\ell_A}(\mathcal{H})\big).$$

As demonstrated by Mao et al. [2024h], Mohri and Zhong [2025], the minimizability gap is always non-negative and is bounded above by the approximation error $\mathcal{A}_\ell(\mathcal{H}) = \mathcal{R}_\ell^*(\mathcal{H}) - \mathcal{R}_\ell^*(\mathcal{H}_{\mathrm{all}})$, i.e., $0 \leq \mathcal{M}_\ell(\mathcal{H}) \leq \mathcal{A}_\ell(\mathcal{H})$. The minimizability gap becomes zero when $\mathcal{H} = \mathcal{H}_{\mathrm{all}}$ or, more generally, when the approximation error $\mathcal{A}_\ell(\mathcal{H}) = 0$. In other cases, it is typically non-zero and offers a more refined measure than the approximation error. In particular, $\mathcal{H}$-consistency bounds imply Bayes-consistency when $\mathcal{H} = \mathcal{H}_{\mathrm{all}}$ and generally provide stronger and more applicable guarantees.

## 5.2 GLA Losses

We now analyze the consistency properties of the GLA loss family. We establish that the LA loss is only Bayes-consistent for $\tau = 1$.

**Bayes-Consistency.** It is known that the Logit-Adjusted (LA) loss is Bayes-consistent with respect to the balanced loss when its temperature parameter is set to one, $\tau = 1$ [Menon et al., 2021]. We begin by establishing a negative result: this consistency does not extend to other values of $\tau$.

**Theorem 2.** *When $\tau \neq 1$, the LA loss $\ell_{\mathrm{LA}}$ is not Bayes-consistent with respect to the balanced loss $\ell_{\mathrm{BAL}}$.*

The proof, which involves characterizing the Bayes classifiers for both the LA loss and the balanced loss, is detailed in Appendix F. In contrast, the following result establishes the Bayes-consistency of the GLA loss with respect to the balanced loss for any $q \in [0, 1)$.

**Theorem 3.** *For any $q \in [0, 1)$, the GLA Loss $\ell_{\mathrm{GLA}}$ is Bayes-consistent with respect to the balanced loss $\ell_{\mathrm{BAL}}$.*

The proof, provided in Appendix G, characterizes the Bayes classifiers for the GLA loss. Note that Theorem 3 recovers the Bayes-consistency of the LA loss (when $q = 0$) as a special case, consistent with [Menon et al., 2021].

$\mathcal{H}$**-Consistency Bounds.** We first present a counter-example (Figure 1) demonstrating that even when

$\tau = 1$ (that is, for the standard LA loss, which is GLA with $q = 0$), $\ell_{\mathrm{LA}}$ is not $\mathcal{H}$-consistent with respect to the balanced loss $\ell_{\mathrm{BAL}}$ for certain bounded hypothesis sets. In this example, considering a two-dimensional distribution where $x_1 \sim U[0,1]$ and $x_2 \mid x_1 \sim \mathcal{N}(yx_1, x_1^2)$, with $y$ following a Bernoulli distribution ($\mathbb{P}(+1) = \frac{1}{8}$), if the hypothesis set consists of linear models with bounded weights, specifically $\{(x,y) \mapsto w_y \cdot x : \|w_y\| = 100\}$, the best-in-class classifier for both the balanced loss and a GCA loss is $x_2 = 0$. However, the best-in-class classifier for the LA loss (with $\tau = 1$) differs and is not parallel to $x_2 = 0$. This implies that the LA loss with $\tau = 1$ is not $\mathcal{H}$-consistent for this bounded hypothesis set.

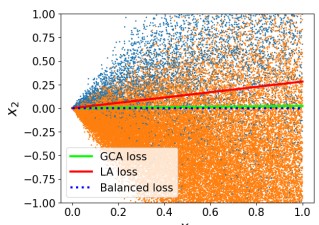

Figure 1: Counterexample to the $\mathcal{H}$-consistency of $\ell_{\mathrm{LA}}$ for bounded hypothesis sets.

This counterexample shows that GLA losses do not guarantee $\mathcal{H}$-consistency for bounded hypothesis sets. The following theorem establishes that the GLA loss $\ell_{\mathrm{GLA}}$ is $\mathcal{H}$-consistent with respect to the balanced loss $\ell_{\mathrm{BAL}}$ if the hypothesis set $\mathcal{H}$ is *complete* that is, for every instance $x \in \mathcal{X}$, the scoring vectors spanned by $\mathcal{H}$ cover the entire space $\mathbb{R}^n$: $\{h(x,\cdot) : h \in \mathcal{H}\} = \mathbb{R}^n$. Naturally, bounded hypothesis sets cannot satisfy this condition. Note that a complete set can be a strict subset of $\mathcal{H}_{\mathrm{all}}$. For example, linear models with unbounded weights are complete, yet they do not equal $\mathcal{H}_{\mathrm{all}}$. Note, the same positive result does not hold for LA losses with general $\tau$s. Being not Bayes-consistent, LA losses are not $\mathcal{H}$-consistent for complete hypothesis sets.

**Theorem 4.** *Assume that $\mathcal{H}$ is complete. Then, for any $q \in [0,1)$, the following $\mathcal{H}$-consistency bound holds for the GLA loss $\ell_{\mathrm{GLA}}$:*

$$\mathcal{R}_{\ell_{\mathrm{BAL}}}(h) - \mathcal{R}^*_{\ell_{\mathrm{BAL}}}(\mathcal{H}) + \mathcal{M}_{\ell_{\mathrm{BAL}}}(\mathcal{H}) \le \Gamma\big(\mathcal{R}_{\ell_{\mathrm{GLA}}}(h) - \mathcal{R}^*_{\ell_{\mathrm{GLA}}}(\mathcal{H}) + \mathcal{M}_{\ell_{\mathrm{GLA}}}(\mathcal{H})\big),$$

*where $\Gamma(t) = \frac{\sqrt{2t}}{\mathsf{p}_{\min}}$ for $q = 0$, and $\Gamma(t) = \frac{\sqrt{2t}}{(\mathsf{p}_{\min})^{\frac{1}{1-q}}(1-q)^{\frac{1}{2}}}$ for $q \in (0,1)$. In the special case where the approximation error $\mathcal{A}_{\ell_{\mathrm{GLA}}}(\mathcal{H}) = 0$, the bound simplifies to:*

$$\mathcal{R}_{\ell_{\mathrm{BAL}}}(h) - \mathcal{R}^*_{\ell_{\mathrm{BAL}}}(\mathcal{H}) \le \Gamma\big(\mathcal{R}_{\ell_{\mathrm{GLA}}}(h) - \mathcal{R}^*_{\ell_{\mathrm{GLA}}}(\mathcal{H})\big),$$

The proof, presented in Appendix H, consists of first defining a Gibbs distribution induced by $h$ and next of applying a Pinsker-type inequality. Our technique is novel: it constructively upper-bounds the conditional regret of the balanced loss by that of the GLA loss, leveraging Lemma 1. Remarkably, when $q = 0$, Theorem 4 yields $\mathcal{H}$-consistency guarantees for the LA loss with $\tau = 1$ under the completeness assumption, a significantly stronger guarantee that the previously established Bayes-consistency result of Menon et al. [2021]. The $\mathcal{H}$-consistency bounds for GLA losses depend inversely on the minimum class probability, scaling as $1/\mathsf{p}_{\min}$ when $q = 0$ and, more generally, as $(1/\mathsf{p}_{\min})^{\frac{1}{1-q}}$ when $q \in (0,1)$,

### 5.3 GCA Losses

This section presents consistency guarantees for GCA losses. We define a hypothesis set $\mathcal{H}$ as *regular* if, for any $x \in \mathcal{X}$, the predictions made by the hypotheses in $\mathcal{H}$ cover the complete set of $n$ possible classification labels: $\mathsf{H}(x) = \{\mathsf{h}(x) : h \in \mathcal{H}\} = [\mathsf{n}]$. Widely used hypothesis sets, such as linear models, neural network families, as well as the family of all measurable functions, are all regular. In particular, every complete hypothesis set is regular, while regularity alone is a much weaker yet natural assumption in practice.

The following theorem shows that for a regular hypothesis set, if a GCE loss $\ell_{\mathrm{GCE}}$ is $\mathcal{H}$-consistent with respect to $\ell_{0-1}$ then its corresponding GCA loss $\ell_{\mathrm{GCA}}$ (Eq. (5)) is also $\mathcal{H}$-consistent with respect to the balanced loss $\ell_{\mathrm{BAL}}$ (Eq. (1)). For simplicity, we assume $\rho_y = 1$ for all $y$ throughout this section.

**Theorem 5.** *Let $\mathcal{H}$ be a regular hypothesis set and $\ell_{\mathrm{GCE}}$ a GCE loss. Assume that there exists a function $\Gamma(t) = \beta t^\alpha$ for some $\alpha \in (0,1]$ and $\beta > 0$, such that the following $\mathcal{H}$-consistency bound holds for all $h \in \mathcal{H}$ and any distribution,*

$$\mathcal{R}_{\ell_{0-1}}(h) - \mathcal{R}^*_{\ell_{0-1}}(\mathcal{H}) + \mathcal{M}_{\ell_{0-1}}(\mathcal{H}) \le \Gamma\big(\mathcal{R}_{\ell_{\mathrm{GCE}}}(h) - \mathcal{R}^*_{\ell_{\mathrm{GCE}}}(\mathcal{H}) + \mathcal{M}_{\ell_{\mathrm{GCE}}}(\mathcal{H})\big).$$

*Then, the following $\mathcal{H}$-consistency bound holds for $\ell_{\mathrm{GCA}}$ with respect to $\ell_{\mathrm{BAL}}$ for all $h \in \mathcal{H}$ and any distribution:*

$$\mathcal{R}_{\ell_{\mathrm{BAL}}}(h) - \mathcal{R}^*_{\ell_{\mathrm{BAL}}}(\mathcal{H}) + \mathcal{M}_{\ell_{\mathrm{BAL}}}(\mathcal{H}) \le \overline{\Gamma}\big(\mathcal{R}_{\ell_{\mathrm{GCA}}}(h) - \mathcal{R}^*_{\ell_{\mathrm{GCA}}}(\mathcal{H}) + \mathcal{M}_{\ell_{\mathrm{GCA}}}(\mathcal{H})\big),$$

where $\overline{\Gamma}(t) = \beta\left(\frac{1}{\mathsf{p}_{\min}}\right)^{1-\alpha} t^\alpha$. *In the special case where the approximation error $\mathcal{A}_{\ell_{\mathrm{GCA}}}(\mathcal{H}) = 0$, this bound simplifies to:*

$$\mathcal{R}_{\ell_{\mathrm{BAL}}}(h) - \mathcal{R}^*_{\ell_{\mathrm{BAL}}}(\mathcal{H}) \leq \overline{\Gamma}\big(\mathcal{R}_{\ell_{\mathrm{GCA}}}(h) - \mathcal{R}^*_{\ell_{\mathrm{GCA}}}(\mathcal{H})\big).$$

The proof is provided in Appendix E, where we constructively define new conditional probabilities $\mathsf{q}(y \mid x)$ along with a normalization factor $Z(x) = \sum_{y \in \mathcal{Y}} \frac{\mathsf{p}(y|x)}{\mathsf{p}(y)} \leq \frac{1}{\mathsf{p}_{\min}}$. These probabilities transform the conditional regret of the balanced loss and the GCA loss into the conditional regrets of the zero-one loss and the GCE loss, respectively, under the newly defined distribution.

When $\mathcal{A}_{\ell_{\mathrm{GCA}}}(\mathcal{H}) = 0$, the $\mathcal{H}$-consistency bound guarantees that if the surrogate estimation error $\mathcal{R}_{\ell_{\mathrm{GCA}}}(h) - \mathcal{R}^*_{\ell_{\mathrm{GCA}}}(\mathcal{H})$ is optimized up to $\epsilon$, the estimation error for the balanced loss, $\mathcal{R}_{\ell_{\mathrm{BAL}}}(h) - \mathcal{R}^*_{\ell_{\mathrm{BAL}}}(\mathcal{H})$, is upper-bounded by $\Gamma(\epsilon)$. For common choices of $\Psi$ in $\ell_{\mathrm{GCA}}$, Mao et al. [2023f,b] show that $\Gamma$ takes specific forms: for $\Psi(t) = -\log(t)$, $\Gamma(t) = \sqrt{2t}$ (so $\alpha = 1/2$ and $\beta = \sqrt{2}$); for $\Psi(t) = \frac{1}{q}(1 - t^q)$ with $q \in (0,1)$, $\Gamma(t) = \sqrt{2n^q t}$ (so $\alpha = 1/2$ and $\beta = \sqrt{2n^q}$). This leads to the following corollary for GCA losses:

**Corollary 6.** *Under the assumptions of Theorem 5, for all $h \in \mathcal{H}$ and any distribution, the following $\mathcal{H}$-consistency bound holds for $\ell_{\mathrm{GCA}}$ with respect to $\ell_{\mathrm{BAL}}$:*

$$\mathcal{R}_{\ell_{\mathrm{BAL}}}(h) - \mathcal{R}^*_{\ell_{\mathrm{BAL}}}(\mathcal{H}) + \mathcal{M}_{\ell_{\mathrm{BAL}}}(\mathcal{H}) \leq \overline{\Gamma}\big(\mathcal{R}_{\ell_{\mathrm{GCA}}}(h) - \mathcal{R}^*_{\ell_{\mathrm{GCA}}}(\mathcal{H}) + \mathcal{M}_{\ell_{\mathrm{GCA}}}(\mathcal{H})\big),$$

*where $\overline{\Gamma}(t) = \frac{\sqrt{2t}}{\sqrt{\mathsf{p}_{\min}}}$ for $\Psi(t) = -\log(t)$ and $\overline{\Gamma}(t) = \frac{\sqrt{2n^q t}}{\sqrt{\mathsf{p}_{\min}}}$ for $\Psi(t) = \frac{1}{q}(1 - t^q)$ with $q \in (0,1)$. In the special case where the approximation error $\mathcal{A}_{\ell_{\mathrm{GCA}}}(\mathcal{H}) = 0$, this bound simplifies to:*

$$\mathcal{R}_{\ell_{\mathrm{BAL}}}(h) - \mathcal{R}^*_{\ell_{\mathrm{BAL}}}(\mathcal{H}) \leq \overline{\Gamma}\big(\mathcal{R}_{\ell_{\mathrm{GCA}}}(h) - \mathcal{R}^*_{\ell_{\mathrm{GCA}}}(\mathcal{H})\big),$$

If $\mathcal{H} = \mathcal{H}_{\mathrm{all}}$, taking the limit on both sides implies the Bayes-consistency of these GCA losses $\ell_{\mathrm{GCA}}$ with respect to the balanced loss $\ell_{\mathrm{BAL}}$. More generally, Corollary 6 demonstrates that $\ell_{\mathrm{GCA}}$ admits an excess error bound relative to $\ell_{\mathrm{BAL}}$ if $\ell_{\mathrm{GCE}}$ has such a bound relative to $\ell_{0-1}$.

Mao et al. [2023f] and Mao et al. [2023b] showed that loss functions belonging to the widely used general cross-entropy (GCE) family (including logistic loss) admit $\mathcal{H}$-consistency bounds with respect to the multi-class zero-one loss $\ell_{0-1}$ when the hypothesis set is complete and *bounded*, respectively. We say a hypothesis set $\mathcal{H}$ is *bounded* if $\mathcal{H} = \{h : \mathcal{X} \times \mathcal{Y} \to \mathbb{R} \mid h(\cdot, y) \in \mathcal{F}, \ \forall y \in \mathcal{Y}\}$, where $\mathcal{F}$ is a family of real-valued functions $f$ satisfying $|f(x)| \leq \Lambda(x)$ for all $x \in \mathcal{X}$, and all values in $[-\Lambda(x), +\Lambda(x)]$ are attainable. Here, $\Lambda(x) > 0$ is a fixed function on $\mathcal{X}$. Boundedness also implies regularity. Thus, a key advantage of GCA losses is their general $\mathcal{H}$-consistency: they are $\mathcal{H}$-consistent for any hypothesis set that is bounded or complete. Furthermore, their consistency bounds exhibit an improved scaling with the minimum class probability, $1/\sqrt{\mathsf{p}_{\min}}$. This contrasts favorably with GLA losses, offering potentially stronger theoretical support in highly imbalanced settings.

**Comparison and Discussion.** Our theoretical analysis reveals distinct characteristics for the two loss families: GLA losses are Bayes-consistent (for $q \in [0,1)$). However, their $\mathcal{H}$-consistency requires the hypothesis set $\mathcal{H}$ to be complete (and thus unbounded). The corresponding bounds depend on the minimum class probability $\mathsf{p}_{\min}$, scaling as $1/\mathsf{p}_{\min}$ (for $q = 0$) or less favorably as $(1/\mathsf{p}_{\min})^{\frac{1}{1-q}}$ ( for $q \in (0,1)$). In contrast, GCA losses demonstrate $\mathcal{H}$-consistency for any hypothesis set that is bounded or complete. Their $\mathcal{H}$-consistency bounds scale more favorably with the minimum class probability, as $1/\sqrt{\mathsf{p}_{\min}}$. This suggests GCA losses offer stronger theoretical guarantees, particularly in settings with significant class imbalance or when using more restricted hypothesis sets.

The trade-offs between these theoretical properties and empirical performance are important. As we will show in the next experimental section (Section 6), GLA losses often achieve slightly better empirical results on common benchmarks. Conversely, GCA losses tend to have an edge in highly imbalanced scenarios. This empirical behavior aligns with our theoretical findings: GLA losses may be preferred for moderately imbalanced scenarios when using expressive, potentially unbounded hypothesis sets where their specific form of logit adjustment is beneficial; GCA losses are theoretically better-suited for highly imbalanced settings due to their favorable consistency scaling and applicability to a wider range of hypothesis sets. For bounded hypothesis sets where GLA's $\mathcal{H}$-consistency is not guaranteed, GCA is the theoretically preferred option.

The assumptions in this section primarily concern properties of the hypothesis set. These are standard and typically satisfied in practice. Most natural hypothesis sets, such as linear models, neural networks, and the set of all measurable functions, are regular, meaning they produce predictions across all $n$ classes. Whether a hypothesis set is bounded or complete depends on the modeling choice (e.g., bounded weights in linear models). Importantly, our results do not assume any specific data distribution and hold for arbitrary distributions, including those arising in real-world settings.

Compared to the previous work [Cortes et al., 2025], the key difference is that IMMAX [Cortes et al., 2025] is designed for optimizing the standard multi-class 0-1 loss under imbalanced data, whereas the proposed GCE and GCA losses are designed to optimize the balanced loss. As a result, IMMAX enjoys consistency with respect to the standard 0-1 loss, while GCE and GCA are consistent with respect to the balanced loss, a property most existing surrogate losses lack, as discussed in Section 3.3.

Appendix B further provides margin bounds for both the GCA and GLA losses in the more general cost-sensitive multi-class classification setting. We show that both losses benefit from margin guarantees, with more favorable bounds for GCA losses, as the GLA bounds depend on $1/\mathsf{p}_{\min}$.

**Theoretical novelty.** Classical margin bounds have been extensively studied (see, for example [Koltchinskii and Panchenko, 2000, 2002, Schapire et al., 1997, Cortes et al., 2021, Mohri et al., 2018]). In particular, Mohri et al. [2018] derived margin bounds for standard multi-class classification. In contrast, we derive new margin bounds for cost-sensitive classification, a setting that introduces additional complexity due to the presence of instance-dependent cost functions. This requires the development of new proof techniques, including the derivation of an upper bound on the loss function expressed in terms of a margin loss and a maximum operator, along with an analysis of the Rademacher complexity of this maximum term via the vector contraction lemma. Moreover, in addition to the resulting margin bounds for GCA loss functions, our margin bounds for GLA loss functions are non-trivial and require a specific and entirely new analysis (Appendix B.2). Mao et al. [2023f,b] studied $\mathcal{H}$-consistency bounds for loss functions in the general cross-entropy (GCE) family with respect to the standard zero-one loss. In contrast, our work establishes $\mathcal{H}$-consistency bounds for the proposed GCA and GLA losses with respect to the balanced loss, where both the surrogate and target losses are more complex. This required several novel technical contributions, including a characterization of the conditional regret of the balanced loss, the use of Gibbs distributions and Pinsker-type inequalities for analyzing GLA losses, and a reduction of the conditional regrets of the balanced and GCA losses to those of the zero-one and GCE losses under a newly defined distribution.

## 6   Experiments

This section details the empirical evaluation of our proposed Generalized Logit-Adjusted (GLA) and Generalized Class-Aware (GCA) loss functions. We compare their effectiveness in minimizing the balanced loss against several baseline methods on the CIFAR-10, CIFAR-100 [Krizhevsky, 2009], and Tiny ImageNet [Le and Yang, 2015] datasets with respectively 10, 100 and 200 classes. To simulate class imbalance, we reduced the percentage of examples per class identically in both training and test sets, following exactly the protocol in [Menon et al., 2021]. Two types of imbalance were considered: Long-tailed imbalance where class sample sizes decrease exponentially across sorted classes [Cui et al., 2019], and Step imbalance where minority classes share one sample size, and majority classes share another, creating a distinct two-group split [Buda et al., 2018]. The severity of imbalance is quantified by the imbalance ratio, $\rho = \frac{\max_{k=1}^{n} m_k}{\min_{k=1}^{n} m_k}$, where $m_k$ is the number of samples in class $k$. We evaluated performance at $\rho = 100$ (C), following Menon et al. [2021], and at a more extreme setting of $\rho = 1000$ (M).

Our experimental setup, including training procedures and neural network architectures, strictly followed Menon et al. [2021]. We used a ResNet-32 architecture with ReLU activations [He et al., 2016]. Standard data augmentation techniques were applied: for CIFAR-10 and CIFAR-100, this involved 4-pixel padding followed by $32 \times 32$ random crops and random horizontal flips; for Tiny ImageNet, 8-pixel padding was used, followed by $64 \times 64$ random crops. All models were trained for 200 epochs using Stochastic Gradient Descent (SGD) with Nesterov momentum [Nesterov, 1983]. We used a a batch size of $1,024$, a weight decay of $1 \times 10^{-3}$, and a cosine decay learning rate schedule [Loshchilov and Hutter, 2016] without restarts, with an initial learning rate of $0.2$.

Table 1: Balanced error of ResNet-32 on *long-tailed* (left) and *step-imbalanced* (right) imbalanced CIFAR-10, CIFAR-100 and Tiny ImageNet; means ± standard deviations over 5 runs. Note, we are reporting total error and not dividing by number of classes. Imbalance ratios $\rho$ = 1000 (M), 100 (C).

| Method | $\rho$ | CIFAR-10 | CIFAR-100 | Tiny I.Net | Method | $\rho$ | CIFAR-10 | CIFAR-100 | Tiny I.Net |
|---|---|---|---|---|---|---|---|---|---|
| CE | | 2.46 ± 0.09 | 38.45 ± 0.37 | 70.23 ± 0.38 | CE | | 6.33 ± 0.01 | 12.47 ± 0.12 | 39.41 ± 0.40 |
| WCE | | 2.52 ± 0.17 | 39.89 ± 0.76 | 75.89 ± 0.67 | WCE | | 6.44 ± 0.02 | 13.66 ± 0.45 | 39.28 ± 0.31 |
| LA ($\tau$ = 1) | | 2.18 ± 0.18 | 35.92 ± 0.47 | 67.17 ± 0.49 | LA ($\tau$ = 1) | | 5.54 ± 0.48 | 11.42 ± 0.33 | 37.44 ± 0.25 |
| EQUAL | | 2.38 ± 0.07 | 37.33 ± 0.36 | 68.44 ± 0.72 | EQUAL | | 5.89 ± 0.24 | 12.24 ± 0.20 | 38.43 ± 0.44 |
| CB | M | 2.58 ± 0.03 | 41.46 ± 0.41 | 80.22 ± 0.59 | CB | M | 6.38 ± 0.01 | 14.96 ± 0.32 | 47.35 ± 0.73 |
| FOCAL | | 2.43 ± 0.10 | 38.02 ± 0.54 | 69.13 ± 0.83 | FOCAL | | 6.35 ± 0.01 | 12.25 ± 0.17 | 39.21 ± 0.31 |
| LDAM | | 2.39 ± 0.08 | 37.39 ± 0.36 | 68.27 ± 0.81 | LDAM | | 6.34 ± 0.01 | 12.30 ± 0.11 | 38.21 ± 0.27 |
| **GCA** | | **2.02 ± 0.15** | **33.17 ± 0.57** | **64.88 ± 0.66** | **GCA** | | **5.35 ± 0.02** | **10.43 ± 0.15** | **36.32 ± 0.32** |
| GLA | | 2.04 ± 0.15 | 33.99 ± 0.52 | 65.57 ± 0.27 | GLA | | 5.39 ± 0.02 | 10.58 ± 0.19 | 36.57 ± 0.35 |
| CE | | 2.72 ± 0.02 | 61.53 ± 0.29 | 106.93 ± 0.89 | CE | | 3.66 ± 0.15 | 60.16 ± 0.09 | 39.68 ± 0.25 |
| WCE | | 2.80 ± 0.08 | 62.20 ± 0.57 | 112.50 ± 0.97 | WCE | | 3.68 ± 0.11 | 61.40 ± 0.51 | 43.68 ± 0.42 |
| LA ($\tau$ = 1) | | 2.23 ± 0.08 | 56.23 ± 0.21 | 102.81 ± 0.89 | LA ($\tau$ = 1) | | 2.70 ± 0.12 | 55.43 ± 0.63 | 38.42 ± 0.14 |
| EQUAL | | 2.60 ± 0.08 | 57.25 ± 0.40 | 104.91 ± 0.84 | EQUAL | | 3.18 ± 0.12 | 57.73 ± 0.54 | 38.91 ± 0.20 |
| CB | C | 2.76 ± 0.04 | 61.55 ± 0.28 | 115.22 ± 0.71 | CB | C | 3.81 ± 0.02 | 66.41 ± 0.11 | 50.51 ± 0.45 |
| FOCAL | | 2.70 ± 0.06 | 61.21 ± 0.24 | 105.47 ± 0.59 | FOCAL | | 3.60 ± 0.11 | 60.06 ± 0.13 | 39.63 ± 0.27 |
| LDAM | | 2.66 ± 0.08 | 60.37 ± 0.60 | 103.99 ± 0.58 | LDAM | | 3.41 ± 0.10 | 58.95 ± 0.11 | 38.67 ± 0.19 |
| GCA | | 2.19 ± 0.08 | 54.02 ± 0.38 | 101.34 ± 0.81 | GCA | | 2.57 ± 0.04 | 53.85 ± 0.47 | 37.59 ± 0.43 |
| **GLA** | | **2.07 ± 0.06** | **53.68 ± 0.76** | **100.70 ± 0.83** | **GLA** | | **2.48 ± 0.11** | **52.70 ± 0.15** | **36.71 ± 0.33** |

We compared our GLA and GCA losses against a suite of widely used baseline methods: standard cross-entropy (CE) loss, class-weighted cross-entropy (WCE) loss [Xie and Manski, 1989, Morik et al., 1999], Logit Adjusted (LA) loss [Menon et al., 2021], Equalization (EQUAL) loss [Tan et al., 2020], Class-Balanced (CB) loss [Cui et al., 2019], FOCAL loss [Ross and Dollár, 2017] and the LDAM loss [Cao et al., 2019]. For all methods, including our GLA and GCA losses, we tune the hyperparameters using a validation set held out separately from the training set. For the parameter $q$ in both GLA and GCA, we selected values from $\{0.0, 0.1, \ldots, 0.9\}$, which are standard choices within the general cross-entropy family. Its performance depends on dataset imbalance (e.g., long-tailed vs. step imbalance). Further details about the experiments including baselines are provided in Appendix C. Performance was primarily evaluated using the balanced error on the imbalanced test sets (i.e., the average of the balanced loss over the test data). Results were averaged over five independent runs, and we report means and standard deviations. Table 1 presents the balanced error for ResNet-32 on long-tailed and step-imbalanced versions of CIFAR-10, CIFAR-100, and Tiny ImageNet.

The results in Table 1 highlight that both our proposed GCA losses and GLA losses generally outperform key baselines such as class-weighted cross-entropy (WCE) and Logit-Adjusted (LA) losses across the tested datasets and imbalance types. This demonstrates the efficacy of our novel loss formulations in achieving better balanced error, indicating improved fairness and accuracy on minority classes. Comparing our two proposed families, GLA losses often achieve the best overall results on several benchmarks, particularly under moderate imbalance ($\rho$ = 100). However, GCA losses in accordance with its better $1/\sqrt{p_{\min}}$ bound tend to exhibit an advantage in settings with high class imbalance ($\rho$ = 1000).

The strong performance of GCA losses, especially their edge in highly imbalanced scenarios ($\rho$ = 1000), underscores the impact of using class-dependent confidence margins. These margins allow GCA to adapt more effectively to severe skews in data distribution compared to simpler weighting or logit adjustment techniques. The performance difference observed between $\rho$ = 100 and $\rho$ = 1000 across all methods, and particularly the relative strengths of GLA and GCA, highlights the sensitivity of these approaches to the severity of class imbalance.

# 7 Conclusion

We introduced two novel families of surrogate losses, GLA and GCA losses, for balanced multi-class classification under class imbalance. Both are principled extensions of widely used loss designs, and our theoretical analysis establishes their consistency properties, highlighting the more favorable $\mathcal{H}$-consistency bounds of GCA losses in imbalanced regimes. Empirically, both loss families outperform existing baselines, with GLA performing better in common benchmarks and GCA offering an edge in highly imbalanced settings. These results position GLA and GCA losses as state-of-the-art surrogates for balanced classification, bridging the gap between fairness, consistency, and practical performance. The extension of these surrogate loss families to structured prediction or multi-label classification could significantly broaden their impact. Finally, refining consistency bounds under realistic hypothesis classes and leveraging recent enhanced $\mathcal{H}$-consistency bounds could provide deeper insights into the behavior of these and related loss functions in balanced learning settings.

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

# Contents of Appendix

# A   Related work

Class imbalance is a prevalent challenge in real-world multi-class classification problems [Cui et al., 2019, Fawcett and Provost, 1996, Kang et al., 2021, Kubat and Matwin, 1997, Lewis and Gale, 1994, Liu et al., 2019, Menon et al., 2021]. Applications such as medical diagnosis, fraud detection, and rare event prediction often involve highly skewed label distributions, where a small subset of classes dominate the data, while others, sometimes the most critical, are heavily underrepresented. Standard training objectives, such as minimizing the unweighted cross-entropy loss, tend to be biased toward majority classes, leading to poor performance on minority classes and undermining the fairness, soundness and reliability of learned models.

The extensive literature on class imbalance has yielded a diverse array of techniques [Cardie and Nowe, 1997, Chawla et al., 2002, He and Garcia, 2009, Kubat and Matwin, 1997, Wallace et al., 2011]. Due to space constraints, a comprehensive review of every method is infeasible. Instead, we will categorize and discuss several major strategic directions, referring the reader to recent surveys, such as Zhang et al. [2023], for a more exhaustive treatment. These strategies can be broadly grouped as follows:

**1.  Data-Level Approaches** These methods aim to directly modify the training dataset's class distribution to create a more balanced representation.

- **Re-sampling Techniques:** This is the most traditional approach, involving either oversampling the minority classes (e.g., by duplicating instances or more advanced interpolation) or undersampling the majority classes (by removing instances) [Kubat and Matwin, 1997, Wallace et al., 2011].

- **Synthetic Data Generation:** More sophisticated methods generate new synthetic samples for minority classes. SMOTE (Synthetic Minority Over-sampling Technique) and its variants are prominent examples [Chawla et al., 2002, Han et al., 2005, Qiao and Liu, 2008].

- **Advanced Data Augmentation:** Recent works explore targeted data augmentation strategies to enhance minority class representation, sometimes using generative models or optimal transport principles (e.g., [Gao et al., 2023, Liu et al., 2024, Wang et al., 2021a, Zhu et al., 2024]). While these methods can improve minority class recognition, oversampling may lead to overfitting, undersampling can discard valuable data, and the effectiveness of synthetic data depends heavily on the generation quality [Estabrooks et al., 2004, Liu et al., 2008, Shi et al., 2023, Zhang and Pfister, 2021].

**2.  Algorithm-Level Cost-Sensitive Learning** This category focuses on modifying the learning algorithm to treat classes differently, typically by assigning higher misclassification costs to errors on minority classes.

- **Class Re-Weighting:** A common implementation involves incorporating class weights directly into the loss function, where weights are often inversely proportional to class frequencies or based on concepts like the "effective number of samples" [Cui et al., 2019]. Examples include weighted versions of Softmax or the 0/1 loss [Gabidolla et al., 2024, Morik et al., 1999, Xie and Manski, 1989].

- **Cost-Sensitive Classifiers:** Some learning algorithms, like SVMs, have explicit cost-sensitive formulations [Iranmehr et al., 2019, Masnadi-Shirazi and Vasconcelos, 2010]. Many other methods adapt standard learners to be cost-aware [Elkan, 2001, Fan et al., 2017, Jamal et al., 2020, Sun et al., 2007, Wang et al., 2022, Suh and Seo, 2023, Wang et al., 2023, 2025, Li et al., 2025, Xiao et al., 2023, Zhang et al., 2018, 2019b, 2022, Zhao et al., 2018, Zhou and Liu, 2005]. Cost-sensitive methods offer a principled way to emphasize underrepresented classes. While they can be viewed as algorithmically achieving effects similar to re-sampling, they avoid explicit data duplication or removal. However, their success often hinges on the appropriate selection of costs/weights, and they may not fundamentally alter the decision boundaries if the classes are inherently hard to separate or if the chosen weights are not optimal [Van Hulse et al., 2007].

**3. Loss Function and Logit Adjustment** This broad category involves designing or modifying loss functions to be more robust to class imbalance or to directly optimize for balanced performance metrics.

- **Modulating Sample Contributions:** Some losses dynamically adjust the contribution of each sample to the total loss based on its difficulty or class. The Focal loss [Lin et al., 2017], for instance, down-weights well-classified (often majority class) examples, allowing the model to focus on hard, minority examples.

- **Margin-Based Modifications:** Several approaches aim to enforce larger decision margins for minority classes or between specific class pairs. Examples include LDAM [Cao et al., 2019], Equalization loss (ESQL) [Tan et al., 2020], and Balanced Softmax [Jiawei et al., 2020]. LADE [Hong et al., 2021] also explores disentangling label distributions.

- **Direct Logit Adjustments:** This sub-group directly modifies the logits (pre-softmax outputs) of the model, often by adding class-specific biases. The Logit Adjustment (LA) method by Menon et al. [2021], Khan et al. [2019] and related techniques like UNO-IC [Tian et al., 2020, Wei et al., 2024] and LSC [Wei et al., 2024] fall here. Menon et al. [2021] showed that a specific form of logit adjustment can achieve Bayes-consistency for the balanced error. Other works explore multiplicative logit modifications [Ye et al., 2020] or combinations of additive and multiplicative changes, like the Vector-Scaling loss [Kini et al., 2021], though multiplicative changes can sometimes be seen as equivalent to input feature re-normalization. To capture how these modified loss functions handle different classes, Wang et al. [2023] proposed a novel technique named data-dependent contraction. Wang et al. [2025] showed that the additive and multiplicative logit modifications essentially correspond to different local calibration assumptions. These methods directly influence the optimization landscape and decision boundaries but may introduce new hyperparameters requiring careful tuning.

**4. Representation Learning for Imbalanced Data** Instead of (or in addition to) modifying data or loss functions, these techniques focus on learning feature representations that are inherently more robust to class imbalance or that better highlight minority class characteristics.

- Examples include OLTR [Liu et al., 2019], PaCo [Cui et al., 2021], DisA [Gao et al., 2024], and other recent methods focused on semantic richness or distribution alignment (e.g., RichSem [Meng et al., 2023], RBL [Meng et al., 2023], WCDAS [Han, 2023]). Learning discriminative and balanced representations is a fundamental goal, and these methods often aim to decouple feature learning from classifier training to some extent.

**5. Decoupled Training and Post-Hoc Adjustments** This strategy involves separating the learning process into stages or applying corrections after an initial model has been trained.

- **Decoupled Training:** Representation learning and classifier training are often performed separately. For example, a model might first be trained with instance-balanced sampling or a standard loss, and then the classifier head is fine-tuned using a class-balanced approach (e.g., Decouple-IB-CRT [Kang et al., 2020], CB-CRT [Kang et al., 2020], SR-CRT [Kang et al., 2020], PB-CRT [Kang et al., 2020], MiSLAS [Zhong et al., 2021]). Weight normalization techniques [Kim and Kim, 2019, Kang et al., 2020, Zhang et al., 2019a] also often fall under this paradigm.

- **Post-Hoc Correction:** These methods adjust the outputs or decision thresholds of a pre-trained classifier to improve performance on imbalanced data, without retraining the model [Collell et al., 2016, Fawcett and Provost, 1996, Zhu et al., 2023]. These approaches offer flexibility and can be applied to existing models, but post-hoc methods may not achieve the same level of performance as methods that incorporate imbalance considerations throughout training.

**6. Ensemble Learning Approaches** Ensemble methods combine multiple classifiers to achieve better predictive performance than any single constituent classifier. For imbalanced learning, ensembles are often constructed by training base learners on different re-sampled versions of the data or by using different cost-sensitive strategies for each member.

- Examples include BBN [Zhou et al., 2020], LFME [Xiang et al., 2020], RIDE [Wang et al., 2021b], ResLT [Cui et al., 2022], SADE [Zhang et al., 2022], and DirMixE [Yang et al., 2024]. Ensembles are often robust but can increase computational expense and reduce model interpretability.

**7. Other Notable Strategies** The field also includes various other specialized techniques:

- **Transfer Learning:** Leveraging knowledge from related tasks or datasets can help, especially for data-scarce minority classes (e.g., SSP [Yang and Xu, 2020]).

- **Specialized Classifier Design:** Some works focus on designing classifier architectures or objective functions specifically robust to long tails or confounding factors (e.g., De-confound [Tang et al., 2020], [Kasarla et al., 2022, Yang et al., 2022], LIFT [Shi et al., 2024], SimPro [Du et al., 2024]).

- **Metric-Focused Optimization:** Recent studies also analyze the asymptotic performance of classifiers under different metrics on imbalanced data [Loffredo et al., 2024], develop size-invariant metrics for specific tasks like salient object detection [Li et al., 2024a], and propose improved Average Precision (AP) losses for the AUPRC metric that are robust to noisy and imbalanced pseudo-labels [Wen et al., 2025]. Information and data augmentation via distillation have also been explored [Li et al., 2024b].

This categorization highlights the multifaceted nature of addressing class imbalance. Our work contributes to the area of loss function and logit adjustment, aiming for theoretically grounded and empirically effective solutions. For further details on the landscape of imbalanced learning, we again refer the reader to comprehensive surveys like Zhang et al. [2023].

# B  Margin bounds

This section provides a margin-based theoretical analysis of cost-sensitive multi-class classification. We derive margin bounds for both the GCA and GLA families. The analysis for the GLA family is more complex, and the resulting bound is generally less favorable, with a dependence on $1/\mathsf{p}_{\min}$.

The proof involves the derivation of an upper bound on the cost-sensitive zero-one loss function expressed in terms of a margin loss and a maximum operator, along with an analysis of the Rademacher complexity of this maximum term via the vector contraction lemma. Moreover, our margin bounds for GLA loss functions are non-trivial and require a specific and entirely new analysis (Appendix B.2).

## B.1  Theoretical analysis

Let $h\colon \mathcal{X}\times[n]\to\mathbb{R}$ be scoring function belonging to the hypothesis set $\mathcal{H}$. We define the cost-sensitive zero-one loss function $\mathsf{L}$ as follows: for all $(h,x,y)\in\mathcal{H}\times\mathcal{X}\times[n]$,

$$\mathsf{L}(h,x,k) = c(x,y)\,1_{\mathsf{h}(x)\neq y},$$

where $c(x,y)$ is a non-negative cost that is upper bounded by $\overline{C}$. Note that $\ell_{\mathrm{BAL}}$ is a special case of $\mathsf{L}$.

**A. Cost-sensitive margin loss functions.** We first introduce new cost-sensitive margin loss functions which will play a central role in our derivation of margin-based guarantees for cost-sensitive learning.

Let $\Phi_\rho\colon u\mapsto\min(1,\max(0,1-u/\rho))$ denote the $\rho$-margin loss function. We can upper-bound the cost-sensitive zero-one loss function $\mathsf{L}$ as follows:

$$\begin{aligned}
\mathsf{L}(h,x,y) &\leq c(x,y)\Phi_\rho(\rho_h(x,y)) \\
&= c(x,y)\Phi_\rho\Big(h(x,y) - \max_{y'\neq y} h(x,y')\Big) \\
&\leq c(x,y)\Phi_\rho(h(x,y) - h(x,\mathsf{h}(x))) \\
&= c(x,y)\max_{y'\in[n]}\{\Phi_\rho(h(x,y) - h(x,y'))\}.
\end{aligned}$$

The second inequality follows from the fact that when $y = \mathsf{h}(x)$ we have $h(x,y) = h(x,\mathsf{h}(x)) \geq \max_{y'\neq y} h(x,y')$. Otherwise, for $y \neq \mathsf{h}(x)$, the runner-up prediction satisfies $\mathrm{argmax}_{y'\neq y} h(x,y') = \mathsf{h}(x)$.

The analysis above motivates the definition of the *cost-sensitive margin loss function* as the function $\mathsf{L}_\rho\colon\mathcal{H}_{\mathrm{all}}\times\mathcal{X}\times[n]\to\mathbb{R}$, defined as follows, for any fixed $\rho > 0$:

$$\mathsf{L}_\rho(h,x,y) = c(x,y)\max_{y'\in[n]}\{\Phi_\rho(h(x,y) - h(x,y'))\}.$$

**B. Margin bounds.** We now establish a general margin-based generalization bound, which serves as the foundation for deriving new algorithms for cost-sensitive classification.

Given a sample $S = (x_1, \ldots, x_m)$ and a hypothesis $h$, the *empirical cost-sensitive margin loss* is defined by $\widehat{\mathcal{R}}_{S,\rho}(h) = \frac{1}{m} \sum_{i=1}^m \mathsf{L}_\rho(h, x_i, y_i)$ and the *empirical GCA loss* is defined by $\widehat{\mathcal{R}}_{S,\ell_{\mathrm{GCA}}}(h) = \frac{1}{m} \sum_{i=1}^m \ell_{\mathrm{GCA}}(h, x_i, y_i)$. The empirical Rademacher complexity of $\mathcal{H}$ for a sample $S$ is defined as:

$$\widehat{\mathfrak{R}}_S(\mathcal{H}) = \frac{1}{m} \mathop{\mathbb{E}}_{\epsilon}\left[\sup_{h \in \mathcal{H}}\left\{\sum_{i=1}^m \sum_{y=1}^n \epsilon_{iy} h(x_i, y)\right\}\right],$$

where $\epsilon = (\epsilon_{iy})_{i,y}$ represents a matrix of independent Rademacher variables $\epsilon_{iy}$s, each uniformly distributed over $\{-1, +1\}$. For any integer $m \geq 1$, the Rademacher complexity of $\mathcal{H}$ is the expectation of $\widehat{\mathfrak{R}}_S(\mathcal{H})$ over all samples $S$ of size $m$: $\mathfrak{R}_m(\mathcal{H}) = \mathbb{E}_{S \sim \mathcal{D}^m}[\widehat{\mathfrak{R}}_S(\mathcal{H})]$.

Using these notions of complexity, we prove the following cost-sensitive margin-based guarantees.

**Theorem 7** (Margin bound for cost-sensitive classification)**.** *Let $\mathcal{H}$ be a family of functions mapping from $\mathcal{X} \times [n]$ to $\mathbb{R}$. Then, for any $\delta > 0$, with probability at least $1 - \delta$, each of the following inequalities holds for all $h \in \mathcal{H}$:*

$$\mathcal{R}_{\mathsf{L}}(h) \leq \widehat{\mathcal{R}}_{S,\rho}(h) + 4\overline{C}\sqrt{2n}\,\mathfrak{R}_m(\mathcal{H}) + \sqrt{\frac{\log\frac{1}{\delta}}{2m}}$$

$$\mathcal{R}_{\mathsf{L}}(h) \leq \widehat{\mathcal{R}}_{S,\rho}(h) + 4\overline{C}\sqrt{2n}\,\widehat{\mathfrak{R}}_S(\mathcal{H}) + 3\sqrt{\frac{\log\frac{2}{\delta}}{2m}}.$$

The proof is included in Appendix I. These bounds can be generalized to hold uniformly for all $\rho \in (0, 1]$, at the cost of additional $\log\log$-terms, using standard proof techniques [Mohri et al., 2018, Theorem 5.9]. As with standard margin bounds, these learning guarantees suggest a trade-off: Increasing $\rho$ reduces the complexity term (second term) but simultaneously increases the empirical cost-sensitive margin loss, $\widehat{\mathcal{R}}_{S,\rho}(h)$ (first term), by imposing stricter confidence margin requirements. Thus, if $h$ maintains a low empirical cost-sensitive margin loss even with a relatively large $\rho$ value, it admits a strong generalization error guarantee. Using the fact that $\mathsf{L}_\rho(h)$ is upper bounded by $\ell_{\mathrm{GCA}}(h/\rho)$, where $c(x, y) = \frac{1}{\mathsf{p}(y)} \leq \frac{1}{\mathsf{p}_{\min}} = \overline{C}$, we derive the margin bounds for GCA losses below.

$$\mathcal{R}_{\ell_{\mathrm{BAL}}}(h) \leq \widehat{\mathcal{R}}_{S,\ell_{\mathrm{GCA}}}(h/\rho) + \frac{4}{\mathsf{p}_{\min}}\sqrt{2n}\,\mathfrak{R}_m(\mathcal{H}) + \sqrt{\frac{\log\frac{1}{\delta}}{2m}}$$

$$\mathcal{R}_{\ell_{\mathrm{BAL}}}(h) \leq \widehat{\mathcal{R}}_{S,\ell_{\mathrm{GCA}}}(h/\rho) + \frac{4}{\mathsf{p}_{\min}}\sqrt{2n}\,\widehat{\mathfrak{R}}_S(\mathcal{H}) + 3\sqrt{\frac{\log\frac{2}{\delta}}{2m}}.$$

## B.2 Margin bounds for GLA lossess

The previous section established margin bounds for GCA losses by leveraging their class-weighted structure. In contrast, deriving analogous bounds for GLA losses is non-trivial due to their different formulation, which involves shifting logits based on class priors. To address this, we will rely on a non-trivial inequality presented in Lemma 8.

Given a sample $S = (x_1, \ldots, x_m)$ and a hypothesis $h$, the *empirical GLA loss* is defined by $\widehat{\mathcal{R}}_{S,\ell_{\mathrm{GLA}}}(h) = \frac{1}{m} \sum_{i=1}^m \ell_{\mathrm{GLA}}(h, x_i, y_i)$. For simplicity, our analysis focuses on the GLA loss with $q = 0$. A similar line of reasoning allows for the extension of this proof to the general case where $q \in (0, 1)$. In our setting of the balanced loss, the costs only depend on $y$ with $c(y) = 1/\mathsf{p}(y)$. Our analysis holds for arbitrary such $y$-dependent costs. Let $c_{\max}$ denote an upper bound $c_{\min}$ a lower bound on the costs. Define $C_{\max} = \frac{c_{\max}}{\log\left[1 + \frac{c_{\min}}{c_{\max}}\right]}$. Then, for any $y, y' \in \mathcal{Y}$, the following holds:

$$\frac{c(y)}{\log\left[1 + \frac{c(y)}{c(y')}\right]} \leq \frac{c(y)}{\log\left[1 + \frac{c_{\min}}{c_{\max}}\right]} \leq C_{\max}.$$

Thus, for any $\rho > 0$ and $y, y' \in \mathcal{Y}$, we have (see illustration in Figure 2 and proof of Lemma 8)

$$c(y)\Phi_\rho(v) \leq C_{\max} \log\left[1 + \frac{c(y)}{c(y')}\exp\left(-\frac{v}{\rho}\right)\right].$$

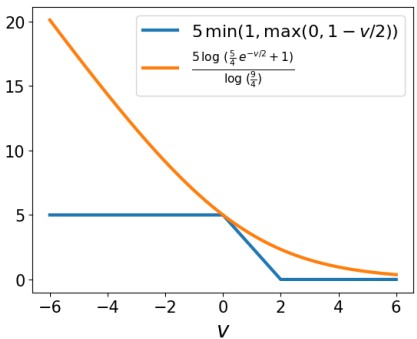

Figure 2: Illustration of the bound in the proof of the margin loss for $\ell_{\mathrm{LA}}$.

Using the monotonicity of the logarithm and upper-bounding a maximum of non-negative terms by a sum yields the following any $\rho > 0$ and any $(x, y) \in \mathcal{X} \times \mathcal{Y}$:

$$
\begin{aligned}
c(y) \max_{y' \neq y} \{\Phi_\rho(h(x,y) - h(x,y'))\} &\leq C_{\max} \max_{y' \neq y} \left\{ \log\left[1 + \frac{c(y)}{c(y')} \exp\left(\frac{h(x,y') - h(x,y)}{\rho}\right)\right]\right\} \\
&\leq C_{\max} \left\{ \log\left[1 + \max_{y' \neq y} \frac{c(y)}{c(y')} \exp\left(\frac{h(x,y') - h(x,y)}{\rho}\right)\right]\right\} \\
&\leq C_{\max} \left\{ \log\left[1 + \sum_{y' \neq y} \frac{c(y)}{c(y')} \exp\left(\frac{h(x,y') - h(x,y)}{\rho}\right)\right]\right\} \\
&= C_{\max} \left\{ \log\left[\sum_{y' \in \mathcal{Y}} \frac{c(y)}{c(y')} \exp\left(\frac{h(x,y') - h(x,y)}{\rho}\right)\right]\right\} \\
&= C_{\max} \ell_{\mathrm{GLA}}(h/\rho, x, y).
\end{aligned}
$$

Thus, this yields the margin-based bounds for the GLA loss below. In our setting, $c_{\min} = 1 \leq \frac{1}{\mathsf{p}(y)}$ and $c_{\max} = \frac{1}{\mathsf{p}_{\min}}$. Thus, as with the $\mathcal{H}$-consistency guarantees, the margin bounds here depend on $\frac{1}{\mathsf{p}_{\min}}$.

$$
\mathcal{R}_{\ell_{\mathrm{BAL}}}(h) \leq \frac{1}{\mathsf{p}_{\min} \log[1 + \mathsf{p}_{\min}]} \widehat{\mathcal{R}}_{S, \ell_{\mathrm{GLA}}}(h/\rho) + \frac{4}{\mathsf{p}_{\min}} \sqrt{2n} \, \mathfrak{R}_m(\mathcal{H}) + \sqrt{\frac{\log \frac{1}{\delta}}{2m}}
$$

$$
\mathcal{R}_{\ell_{\mathrm{BAL}}}(h) \leq \frac{1}{\mathsf{p}_{\min} \log[1 + \mathsf{p}_{\min}]} \widehat{\mathcal{R}}_{S, \ell_{\mathrm{GLA}}}(h/\rho) + \frac{4}{\mathsf{p}_{\min}} \sqrt{2n} \, \widehat{\mathfrak{R}}_S(\mathcal{H}) + 3\sqrt{\frac{\log \frac{2}{\delta}}{2m}}.
$$

**Lemma 8.** *For any $\rho > 0$ and $y, y' \in \mathcal{Y}$, we have*

$$
c(y)\Phi_\rho(v) \leq C_{\max} \log\left[1 + \frac{c(y)}{c(y')} \exp\left(-\frac{v}{\rho}\right)\right],
$$

*for every $v \in \mathbb{R}$.*

*Proof.* Fix labels $y, y'$ and a margin value $v \in \mathbb{R}$. Write $a = \frac{c(y)}{c(y')}$, $t = \frac{v}{\rho}$, and recall the bounds $c_{\min} \leq c(y), c(y') \leq c_{\max}$. By definition, $C_{\max} = \frac{c_{\max}}{\log\left[1 + \frac{c_{\min}}{c_{\max}}\right]}$. Then, for any $y, y' \in \mathcal{Y}$, the following holds:

$$
\frac{c(y)}{\log\left[1 + \frac{c(y)}{c(y')}\right]} \leq \frac{c(y)}{\log\left[1 + \frac{c_{\min}}{c_{\max}}\right]} \leq C_{\max}. \tag{6}
$$

Next, we analyze case by case.

*(i) $v \leq 0$ ($t \leq 0$).* Then $\Phi_\rho(t) = 1$ and $\exp(-t) \geq 1$, using (6) gives

$$
C_{\max} \log\left(1 + ae^{-t}\right) \geq C_{\max} \log(1 + a) \geq c(y) = c(y)\Phi_\rho(t).
$$

*(ii)* $0 \le v \le \rho$ $(0 \le t \le 1)$. Define $h(t) = \log(1 + ae^{-t}) - (1 - t)\log(1 + a)$. Since $h'(t) = -\dfrac{ae^{-t}}{1 + ae^{-t}} + \log(1 + a) \ge -\dfrac{a}{1 + a} + \log(1 + a) \ge 0$ (because $\log(1 + u) \ge u/(1 + u)$ for all $u \ge 0$), we have $h(t) \ge h(0) = 0$; hence

$$(1 - t)\log(1 + a) \le \log\big(1 + ae^{-t}\big).$$

Multiplying by $C_{\max}$ and using (6) gives

$$C_{\max}\log\big(1 + ae^{-t}\big) \ge C_{\max}\log(1 + a)(1 - t) \ge c(y)(1 - t) = c(y)\Phi_\rho(v).$$

*(iii)* $v \ge \rho$ $(t \ge 1)$. Then $\Phi_\rho(v) = 0$ and the desired inequality is trivial because the right–hand side is non-negative.

In conclusion, all three cases yield

$$c(y)\Phi_\rho(v) \le C_{\max}\log\left[1 + \frac{c(y)}{c(y')}\exp\left(-\frac{v}{\rho}\right)\right],$$

for every $v \in \mathbb{R}$. This completes the proof. $\qquad\square$

### B.3 Algorithms

The margin guarantees established in the previous section provide a foundation for developing new algorithms. We begin by deriving a more explicit learning guarantee within a broad framework, which we then use to define a general cost-sensitive learning algorithm.

**A. Explicit upper bounds**. To make these guarantees more explicit, we introduce the following setup. Given a feature mapping $\Phi\colon \mathcal{X} \times [n] \to \mathbb{R}^d$, we can identify $\mathcal{X} \times [n]$ with a subset of $\mathbb{R}^d$, with $\|\Psi(x,y)\| \le \mathsf{X}_y$ for all $x \in \mathcal{X}$ and $\mathsf{X} = \max_{y\in[n]}\mathsf{X}_y$, for some norm $\|\cdot\|$. We assume $\mathcal{H}$ is given by $\mathcal{H} = \big\{h \in \overline{\mathcal{H}}\colon \|h\|_* \le \overline{\mathsf{H}}\big\}$, for some appropriate norm $\|\cdot\|_*$ on some space $\overline{\mathcal{H}}$ and $\overline{\mathsf{H}} > 0$. This formulation covers a wide range of hypothesis sets, including linear, kernel-based, and neural network models. In particular, it captures the settings of neural networks with weight matrices constrained by a Frobenius norm bound [Cortes et al., 2017, Neyshabur et al., 2015] or a spectral norm complexity constraint relative to reference weight matrices [Bartlett et al., 2017]. In all of these cases, the empirical Rademacher complexity can be upper bounded as follows:

$$\widehat{\mathfrak{R}}_S(\mathcal{H}) \le \frac{\sqrt{n}\,\mathsf{H}}{m}\sqrt{\sum_{j=1}^{n}m_j\mathsf{X}_j^2} \le \frac{\sqrt{n}\,\mathsf{H}\mathsf{X}}{\sqrt{m}},$$

where the complexity term $\mathsf{H}$ depends on $\overline{\mathsf{H}}$. Combining this upper bound with Theorem 7 yields the following more explicit guarantee.

**Corollary 9.** *Fix $\rho = [\rho_k]_{k\in[n]}$, then, for any $\delta > 0$, with probability at least $1 - \delta$ over the choice of a sample $S$ of size $m$, the following holds for any $f \in \mathcal{H}$:*

$$\mathcal{R}_\mathsf{L}(h) \le \widehat{\mathfrak{R}}_{S,\rho}(h) + \frac{4\overline{C}\sqrt{2}n\mathsf{H}}{m}\sqrt{\sum_{j=1}^{n}m_j\mathsf{X}_j^2} + 3\sqrt{\frac{\log\frac{2}{\delta}}{2m}}.$$

As with Theorem 7, this bound can be generalized to hold uniformly for all $\rho \in (0,1]$, at the cost of additional $\log\log$-terms. This generalized guarantee provides a basis for designing algorithms choosing $h \in \mathcal{H}$ and $\rho$ to minimize the bound.

Let $\Psi$ be a decreasing convex function such that $\Phi_\rho(x) \le \Psi\left(\frac{x}{\rho}\right)$ for all $x \in \mathbb{R}$ and $\rho > 0$. $\Psi$ may be the hinge loss, $\Psi(x) = \max(0, 1 - x)$, or any member of the broad family of composition-sum (comp-sum) losses [Mao et al., 2023f] defined by $\Psi(x) = \Phi^\tau(e^{-x})$, with $\Phi^\tau$ for $\tau \ge 0$ given by

$$\Phi^\tau(u) = \begin{cases} \frac{1}{1-\tau}\big((1 + u)^{1-\tau} - 1\big) & \tau \ge 0, \tau \ne 1 \\ \log(1 + u) & \tau = 1, \end{cases}$$

for all $u \ge 0$. This family includes the logistic loss ($\tau = 1$) and the exponential loss ($\tau = 0$). Using the fact that $\Phi_\rho(t) \le \Psi\left(\frac{t}{\rho}\right)$, the guarantee of Corollary 9 and its generalization to a uniform bound

can be expressed as: for any $\delta > 0$, with probability at least $1 - \delta$, for all $h \in \mathcal{H}$, where the last term accounts for the $\log$-$\log$ terms and the $\delta$-confidence term

$$\mathcal{R}_{\mathsf{L}}(h) \leq \frac{1}{m}\left[\sum_{i=1}^{m} c(x_i, y_i) \max_{y' \in [n]}\left\{\Psi\left(\frac{h(x_i, y_i) - h(x_i, y')}{\rho}\right)\right\}\right] + \frac{4\overline{C}\sqrt{2}n\mathsf{H}}{m}\sqrt{\sum_{j=1}^{n} m_j \mathsf{X}_j^2} + O\left(\frac{1}{\sqrt{m}}\right).$$

**B. General cost-sensitive algorithm.** The bound leads to the following regularization-based algorithm:

$$\min_{h \in \mathcal{H}} \lambda\|h\|^2 + \frac{1}{m}\left[\sum_{i=1}^{m} c(x_i, y_i) \max_{y' \in [n]}\left\{\Psi\left(\frac{h(x_i, y_i) - h(x_i, y')}{\rho}\right)\right\}\right],$$

where $\lambda$ and $\rho$s are selected via cross-validation. This is equivalent to minimizing the following surrogate loss:

$$\ell(h, x, y) = c(x, y) \max_{y' \in [n]}\left\{\Psi\left(\frac{h(x, y) - h(x, y')}{\rho}\right)\right\} \tag{7}$$

The preceding derivation shows that this form can be further upper-bounded by both GCA and GLA losses. Consequently, both loss families benefit from margin guarantees, though GCA losses achieve more favorable bounds due to the GLA bounds' dependence on $1/\mathsf{p}_{\min}$.

## C  Experimental details

This appendix provides supplementary information regarding the experimental setup discussed in Section 6. We first present the precise mathematical formulations for our algorithms and all baseline loss functions used in the comparative analysis. Then, we detail the specific hyperparameter ranges explored during the cross-validation process for each method.

Since our work focuses on principled surrogate losses for imbalanced data, our experiments aimed for a direct comparison with existing losses in their basic forms. We excluded common enhancements from data modification or optimization strategies to isolate the performance of the loss functions.

### C.1  Loss function formulations

Let $m_k$ be the number of samples in class $k$, and $m$ be the total number of samples. Below are the definitions of the loss functions optimized by our algorithms and those optimized by the baseline methods. For any triplet $(h, x, y)$, where $h$ is the hypothesis, $x$ is the input, and $y$ is the true label from a set of $n$ classes:

- **Cross-Entropy (CE) Loss:**

$$\ell_{\text{CE}}(h, x, y) = -\log\left(\frac{e^{h(x,y)}}{\sum_{j=1}^{n} e^{h(x,j)}}\right).$$

- **Class-Weighted Cross-Entropy (WCE) loss** [Morik et al., 1999, Xie and Manski, 1989]:

$$\ell_{\text{WCE}}(h, x, y) = -\frac{m}{m_y}\log\left(\frac{e^{h(x,y)}}{\sum_{j=1}^{n} e^{h(x,j)}}\right).$$

- **Logit Adjusted (LA) Loss** ($\tau = 1$) [Menon et al., 2021]:

$$\ell_{\text{LA}}(h, x, y) = -\log\left(\frac{e^{h(x,y)+\log(m_y)}}{\sum_{j=1}^{n} e^{h(x,j)+\log(m_j)}}\right).$$

- **Equalization (EQUAL) Loss** [Tan et al., 2020]:

$$\ell_{\text{EQUAL}}(h, x, y) = -\log\left(\frac{e^{h(x,y)}}{\sum_{j=1}^{n} w_j e^{h(x,j)}}\right),$$

  with weight $w_j = 1 - \beta \mathbf{1}_{\left\{\frac{m_j}{m} < \lambda\right\}} \mathbf{1}_{\{y \neq j\}}$, where $\beta \sim \text{Bernoulli}(p)$, and $1 > p > 0$, $1 > \lambda > 0$ are hyperparameters.

- **Class-Balanced (CB) Loss** [Cui et al., 2019]:

$$\ell_{\text{CB}}(h, x, y) = -\frac{1 - \gamma}{1 - \gamma^{\frac{m_y}{m}}} \log\left(\frac{e^{h(x,y)}}{\sum_{j=1}^{n} e^{h(x,j)}}\right),$$

where $1 > \gamma > 0$ is a hyperparameter.

- **FOCAL Loss** [Ross and Dollár, 2017]:

$$\ell_{\text{FOCAL}}(h, x, y) = -\left(1 - \frac{e^{h(x,y)}}{\sum_{j=1}^{n} e^{h(x,j)}}\right)^{\gamma} \log\left(\frac{e^{h(x,y)}}{\sum_{j=1}^{n} e^{h(x,j)}}\right),$$

where $\gamma \geq 0$ is a hyperparameter.

- **LDAM Loss** [Cao et al., 2019]:

$$\ell_{\text{LDAM}}(h, x, y) = -\log\left(\frac{e^{h(x,y) - \Delta_y}}{e^{h(x,y) - \Delta_y} + \sum_{j \neq y} e^{h(x,j)}}\right),$$

where $\Delta_j = \frac{C}{m_j^{\frac{1}{4}}}$ for $j \in [n]$, and $C > 0$ is a hyperparameter.

- **Generalized Class-Aware (GCA) Loss**:

$$\ell_{\text{GCA}}(h, x, y) = \frac{m}{m_y} \Psi^q\left(\frac{e^{h(x,y)/\rho_y}}{\sum_{y' \in \mathcal{Y}} e^{h(x,y')/\rho_y}}\right),$$

where $q \in [0, 1)$ and $\boldsymbol{\rho} = (\rho_1, \ldots, \rho_n)$ is a vector of positive parameters for each class.

- **Generalized Logit-Adjusted (GLA) Loss**:

$$\ell_{\text{GLA}}(h, x, y) = \Psi^q\left(\frac{e^{h(x,y) + \frac{\log(m_y/m)}{1-q}}}{\sum_{y' \in \mathcal{Y}} e^{h(x,y') + \frac{\log(m_{y'}/m)}{1-q}}}\right),$$

where $q \in [0, 1)$ is a hyperparameter.

## C.2 Hyperparameter search protocol

As stated in Section 6, all hyperparameters for the baseline methods and our algorithms were optimized via cross-validation. The search ranges for each tunable parameter were as follows:

- **CE Loss, WCE Loss:** These methods do not have tunable hyperparameters beyond standard optimization settings.

- **LA Loss:** We fixed the hyperparameter $\tau = 1$ as the algorithm is only Bayes-consistent for that value.

- **EQUAL Loss:** $p$ was selected from $\{0.1, 0.2, \ldots, 0.9\}$, and $\lambda$ was selected from $\{0.176, 0.5, 0.8, 1.5, 1.76, 2.0, 3.0, 5.0\} \times 10^{-3}$ by following Tan et al. [2020],

- **CB Loss:** $\gamma$ was selected from $\{0.1, 0.2, \ldots, 0.9, 0.99, 0.999, 0.9999\}$ by following Cui et al. [2019],

- **FOCAL Loss:** $\gamma$ was selected from $\{1.0, 1.5, \ldots, 10.0\}$ and $\{0.0, 0.1, \ldots, 0.9\}$ by following Ross and Dollár [2017].

- **LDAM Loss:** $C$ was selected from $\{10^{-4}, \ldots, 10^4\}$ and $\{5 \times 10^{-4}, \ldots, 5 \times 10^3\}$ by following Cao et al. [2019].

- **GCA Loss:** $\boldsymbol{\rho}$ was chosen as $\left(\frac{m_1^{1/3}}{\sum_{k=1}^{n} m_k^{1/3}}, \ldots, \frac{m_n^{1/3}}{\sum_{k=1}^{n} m_k^{1/3}}\right)$ by following Cortes et al. [2025]. $q$ was selected from $\{0.0, 0.1, \ldots, 0.9\}$.

- **GLA Loss:** $q$ was selected from $\{0.0, 0.1, \ldots, 0.9\}$.

# D    Conditional regret for the balanced loss: proof of Lemma 1

**Lemma 1.** *For any $x \in \mathcal{X}$, the best-in-class conditional error and the conditional regret for $\ell_{\mathrm{BAL}}$ can be expressed as follows:*

$$\mathcal{C}^*_{\ell_{\mathrm{BAL}}}(\mathcal{H}, x) = \sum_{y \in \mathcal{Y}} \frac{\mathsf{p}(y \mid x)}{\mathsf{p}(y)} - \max_{y \in \mathsf{H}(\mathsf{x})} \frac{\mathsf{p}(y \mid x)}{\mathsf{p}(y)} \quad \Delta\mathcal{C}_{\ell_{\mathrm{BAL}}, \mathcal{H}}(h, x) = \max_{y \in \mathsf{H}(\mathsf{x})} \frac{\mathsf{p}(y \mid x)}{\mathsf{p}(y)} - \frac{\mathsf{p}(\mathsf{h}(x)) \mid x)}{\mathsf{p}(\mathsf{h}(x))}.$$

*Proof.* By the definition and Bayes' theorem, the conditional error can be expressed as follows:

$$\mathcal{C}_{\ell_{\mathrm{BAL}}}(h, x) = \sum_{y \in \mathcal{Y}} \frac{\mathsf{p}(y \mid x)}{\mathsf{p}(y)} 1_{h(x) \neq y}$$

$$= \sum_{y \in \mathcal{Y}} \frac{\mathsf{p}(y \mid x)}{\mathsf{p}(y)} - \frac{\mathsf{p}(\mathsf{h}(x) \mid x)}{\mathsf{p}(\mathsf{h}(x))}.$$

Since $\{\mathsf{h}(x) \colon h \in \mathcal{H}\} = \mathsf{H}(\mathsf{x})$, the best-in-class conditional error can be expressed as follows:

$$\mathcal{C}^*_{\ell_{\mathrm{BAL}}}(\mathcal{H}, x) = \sum_{y \in \mathcal{Y}} \frac{\mathsf{p}(y \mid x)}{\mathsf{p}(y)} - \max_{y \in \mathsf{H}(\mathsf{x})} \frac{\mathsf{p}(y \mid x)}{\mathsf{p}(y)},$$

which proves the first part of the lemma. This leads to

$$\Delta\mathcal{C}_{\ell_{\mathrm{BAL}}, \mathcal{H}}(h, x) = \mathcal{C}_{\ell_{\mathrm{BAL}}}(h, x) - \mathcal{C}^*_{\ell_{\mathrm{BAL}}}(\mathcal{H}, x) = \max_{y \in \mathsf{H}(\mathsf{x})} \frac{\mathsf{p}(y \mid x)}{\mathsf{p}(y)} - \frac{\mathsf{p}(\mathsf{h}(x) \mid x)}{\mathsf{p}(\mathsf{h}(x))},$$

which proves the second part of the lemma.    $\square$

# E    $\mathcal{H}$-Consistency for the GCA losses: proof of Theorem 5

**Theorem 5.** *Let $\mathcal{H}$ be a regular hypothesis set and $\ell_{\mathrm{GCE}}$ a GCE loss. Assume that there exists a function $\Gamma(t) = \beta t^\alpha$ for some $\alpha \in (0, 1]$ and $\beta > 0$, such that the following $\mathcal{H}$-consistency bound holds for all $h \in \mathcal{H}$ and any distribution,*

$$\mathcal{R}_{\ell_{0-1}}(h) - \mathcal{R}^*_{\ell_{0-1}}(\mathcal{H}) + \mathcal{M}_{\ell_{0-1}}(\mathcal{H}) \leq \Gamma\big(\mathcal{R}_{\ell_{\mathrm{GCE}}}(h) - \mathcal{R}^*_{\ell_{\mathrm{GCE}}}(\mathcal{H}) + \mathcal{M}_{\ell_{\mathrm{GCE}}}(\mathcal{H})\big).$$

*Then, the following $\mathcal{H}$-consistency bound holds for $\ell_{\mathrm{GCA}}$ with respect to $\ell_{\mathrm{BAL}}$ for all $h \in \mathcal{H}$ and any distribution:*

$$\mathcal{R}_{\ell_{\mathrm{BAL}}}(h) - \mathcal{R}^*_{\ell_{\mathrm{BAL}}}(\mathcal{H}) + \mathcal{M}_{\ell_{\mathrm{BAL}}}(\mathcal{H}) \leq \overline{\Gamma}\big(\mathcal{R}_{\ell_{\mathrm{GCA}}}(h) - \mathcal{R}^*_{\ell_{\mathrm{GCA}}}(\mathcal{H}) + \mathcal{M}_{\ell_{\mathrm{GCA}}}(\mathcal{H})\big),$$

*where $\overline{\Gamma}(t) = \beta \left(\frac{1}{\mathsf{p}_{\min}}\right)^{1-\alpha} t^\alpha$. In the special case where the approximation error $\mathcal{A}_{\ell_{\mathrm{GCA}}}(\mathcal{H}) = 0$, this bound simplifies to:*

$$\mathcal{R}_{\ell_{\mathrm{BAL}}}(h) - \mathcal{R}^*_{\ell_{\mathrm{BAL}}}(\mathcal{H}) \leq \overline{\Gamma}\big(\mathcal{R}_{\ell_{\mathrm{GCA}}}(h) - \mathcal{R}^*_{\ell_{\mathrm{GCA}}}(\mathcal{H})\big).$$

*Proof.* The proof involves a reduction of the conditional regrets of the balanced and GCA losses to those of the zero-one and GCE losses under a newly defined distribution and the use of known $\mathcal{H}$-consistency bounds for GCE losses. We define a new conditional probability $\mathsf{q}(y \mid x)$ as $\mathsf{q}(y \mid x) = \frac{\mathsf{p}(y|x)}{\mathsf{p}(y)} \frac{1}{Z(x)}$, where $Z(x) = \sum_{y \in \mathcal{Y}} \frac{\mathsf{p}(y|x)}{\mathsf{p}(y)} \leq \frac{1}{\mathsf{p}_{\min}}$ is the normalization factor. By Lemma 1, the

conditional regret of $\ell_{\mathrm{BAL}}$ can be expressed and upper-bounded as follows:

$$
\begin{aligned}
\Delta \mathcal{C}_{\ell_{\mathrm{BAL}},\mathcal{H}}(h,x) &= \max_{y \in \mathcal{Y}} \frac{\mathsf{p}(y \mid x)}{\mathsf{p}(y)} - \frac{\mathsf{p}(\mathsf{h}(x)) \mid x)}{\mathsf{p}(\mathsf{h}(x))} \\
&= Z(x)\Big( \max_{y \in \mathcal{Y}} \mathsf{q}(y \mid x) - \mathsf{q}(x \mid \mathsf{h}(x)) \Big) \\
&= Z(x)\Delta \mathcal{C}_{\ell_{0-1},\mathcal{H}}(h,x) \\
&\leq Z(x)\Gamma(\Delta \mathcal{C}_{\ell_{\mathrm{GCE}},\mathcal{H}}(h,x)) \qquad\qquad (\mathcal{H}\text{-consistency bound of } \ell_{\mathrm{GCE}}) \\
&= Z(x)\Gamma\Bigg( \sum_{y \in \mathcal{Y}} \mathsf{q}(y \mid x)\ell_{\mathrm{GCE}}(h,x,y) - \inf_{h \in \mathcal{Y}} \sum_{y \in \mathcal{Y}} \mathsf{q}(y \mid x)\ell_{\mathrm{GCE}}(h,x,y) \Bigg) \\
&= Z(x)\Gamma\Bigg( \frac{1}{Z(x)}\Bigg( \sum_{y \in \mathcal{Y}} \frac{\mathsf{p}(y \mid x)}{\mathsf{p}(y)}\ell_{\mathrm{GCE}}(h,x,y) - \inf_{h \in \mathcal{Y}} \sum_{y \in \mathcal{Y}} \frac{\mathsf{p}(y \mid x)}{\mathsf{p}(y)}\ell_{\mathrm{GCE}}(h,x,y) \Bigg) \Bigg) \\
&= Z(x)\Gamma\Bigg( \frac{1}{Z(x)}\Bigg( \sum_{y \in \mathcal{Y}} \mathsf{p}(y \mid x)\ell_{\mathrm{GCA}}(h,x,y) - \inf_{h \in \mathcal{Y}} \sum_{y \in \mathcal{Y}} \mathsf{p}(y \mid x)\ell_{\mathrm{GCA}}(h,x,y) \Bigg) \Bigg) \\
&= Z(x)\Gamma\Bigg( \frac{1}{Z(x)} \Delta \mathcal{C}_{\ell_{\mathrm{GCA}},\mathcal{H}}(h,x) \Bigg) \\
&= \beta\, Z(x)^{1-\alpha} \Delta \mathcal{C}_{\ell_{\mathrm{GCA}},\mathcal{H}}(h,x)^{\alpha} \\
&\leq \beta \left( \frac{1}{p_{\min}} \right)^{1-\alpha} \Delta \mathcal{C}_{\ell_{\mathrm{GCA}},\mathcal{H}}(h,x)^{\alpha}
\end{aligned}
$$

Thus, taking expectations gives:

$$
\begin{aligned}
\mathcal{R}_{\ell_{\mathrm{BAL}}}(h) - \mathcal{R}^*_{\ell_{\mathrm{BAL}}}(\mathcal{H}) + \mathcal{M}_{\ell_{\mathrm{BAL}}}(\mathcal{H}) &= \mathop{\mathbb{E}}_{x}[\Delta \mathcal{C}_{\ell_{\mathrm{BAL}},\mathcal{H}}(h,x)] \\
&\leq \mathop{\mathbb{E}}_{x}\big[ \overline{\Gamma}(\Delta \mathcal{C}_{\ell_{\mathrm{GCA}},\mathcal{H}}(h,x)) \big] \\
&\leq \overline{\Gamma}\Big( \mathop{\mathbb{E}}_{x}[\Delta \mathcal{C}_{\ell_{\mathrm{GCA}},\mathcal{H}}(h,x)] \Big) \\
&\qquad\qquad \text{(concavity of } \Gamma \text{ and Jensen's ineq.)} \\
&= \Gamma\big( \mathcal{R}_{\ell_{\mathrm{GCA}}}(h) - \mathcal{R}^*_{\ell_{\mathrm{GCA}}}(\mathcal{H}) + \mathcal{M}_{\ell_{\mathrm{GCA}}}(\mathcal{H}) \big),
\end{aligned}
$$

where $\overline{\Gamma}(t) = \beta \left( \frac{1}{\mathsf{p}_{\min}} \right)^{1-\alpha} t^{\alpha}$. This concludes the first part of the proof. The second part follows directly from the fact that the minimizability gap $\mathcal{M}_{\ell_{\mathrm{GCA}}}(\mathcal{H})$ vanishes when the approximation error, $\mathcal{A}_{\ell_{\mathrm{GCA}}}(\mathcal{H})$, is zero. This concludes the first part of the proof. The second part follows directly using the fact that when the approximation error is zero: $\mathcal{A}_{\ell_{\mathrm{GCA}}}(\mathcal{H}) = 0$, the minimizability gap $\mathcal{M}_{\ell_{\mathrm{GCA}}}(\mathcal{H})$ vanishes. $\qquad\square$

Note that, for simplicity, we assumed $\rho_y = 1$ for all $y$ in Theorem 5 and its proof. To handle varying values of $\rho_y$, we can directly extend the $\mathcal{H}$-consistency bounds for the general cross-entropy (GCE) family, as derived in [Mao et al., 2023f,b], to the setting where GCE uses distinct $\rho_y$ values. We can then similarly show that these extended bounds for the GCE family can be transformed into bounds for the GCA losses.

# F Negative results for the LA losses: proof of Theorem 2

**Theorem 2.** *When $\tau \neq 1$, the LA loss $\ell_{\mathrm{LA}}$ is not Bayes-consistent with respect to the balanced loss $\ell_{\mathrm{BAL}}$.*

*Proof.* The Bayes classifier $h^*_{\mathrm{LA}}$ of the LA loss satisfies the following condition:

$$
\frac{e^{h^*_{\mathrm{LA}}(x,y)+\tau \log(\mathsf{p}(y))}}{\sum_{y' \in \mathcal{Y}} e^{h^*_{\mathrm{LA}}(x,y')+\tau \log(\mathsf{p}(y'))}} = \mathsf{p}(y \mid x)
$$

By rearranging the terms, we have

$$
\begin{aligned}
e^{h^*_{\mathrm{LA}}(x,y)} &= \frac{\mathsf{p}(y\mid x)}{\mathsf{p}(y)^\tau}\sum_{y'\in\mathcal{Y}}e^{h^*_{\mathrm{LA}}(x,y')+\tau\log(\mathsf{p}(y'))}\\
&= \frac{\mathsf{p}(x\mid y)\mathsf{p}(y)}{\mathsf{p}(y)^\tau\mathsf{p}(x)}\sum_{y'\in\mathcal{Y}}e^{h^*_{\mathrm{LA}}(x,y')+\tau\log(\mathsf{p}(y'))} && \text{(Bayes' theorem)}\\
&= \frac{\mathsf{p}(x\mid y)\mathsf{p}(y)^{1-\tau}}{\mathsf{p}(x)}\sum_{y'\in\mathcal{Y}}e^{h^*_{\mathrm{LA}}(x,y')+\tau\log(\mathsf{p}(y'))}.
\end{aligned}
$$

Thus, since the term $\frac{\sum_{y'\in\mathcal{Y}}e^{h^*_{\mathrm{LA}}(x,y')+\tau\log(\mathsf{p}(y'))}}{\mathsf{p}(x)}$ does not depend on $y$, we obtain

$$
\mathsf{h}^*_{\mathrm{LA}}(x) = \underset{y\in\mathcal{Y}}{\arg\max}\, h^*_{\mathrm{LA}}(x,y) = \underset{y\in\mathcal{Y}}{\arg\max}\, e^{h^*_{\mathrm{LA}}(x,y)} = \underset{y\in\mathcal{Y}}{\arg\max}\, \mathsf{p}(x\mid y)\mathsf{p}(y)^{1-\tau}.
$$

By Lemma 1, we know that the Bayes classifier $h^*_{\mathrm{bal}}$ of the Balanced loss satisfies that

$$
\mathsf{h}^*_{\mathrm{bal}} = \underset{y\in\mathcal{Y}}{\arg\max}\,\frac{\mathsf{p}(y\mid x)}{\mathsf{p}(y)} = \underset{y\in\mathcal{Y}}{\arg\max}\,\mathsf{p}(x\mid y).
$$

Therefore, for any $\tau\neq 1$, there exists a distribution such that $\mathsf{h}^*_{\mathrm{LA}}(x)\neq\mathsf{h}^*_{\mathrm{bal}}$. This implies that when $\tau\neq 1$, the LA loss $\ell_{\mathrm{LA}}$ is not Bayes-consistent with respect to the balanced loss $\ell_{\mathrm{BAL}}$. $\qquad\square$

## G   Bayes-Consistency for the GLA losses: proof of Theorem 3

**Theorem 3.** *For any $q\in[0,1)$, the GLA Loss $\ell_{\mathrm{GLA}}$ is Bayes-consistent with respect to the balanced loss $\ell_{\mathrm{BAL}}$.*

*Proof.* The Bayes classifier $h^*_{\mathrm{GLA}}$ of the GLA loss satisfies the following condition:

$$
\frac{e^{h^*_{\mathrm{GLA}}(x,y)+\frac{\log(\mathsf{p}(y))}{1-q}}}{\sum_{y'\in\mathcal{Y}}e^{h^*_{\mathrm{GLA}}(x,y')+\frac{\log(\mathsf{p}(y'))}{1-q}}} = \frac{(\mathsf{p}(y\mid x))^{\frac{1}{1-q}}}{\sum_{y'\in\mathcal{Y}}(\mathsf{p}(y'\mid x))^{\frac{1}{1-q}}}
$$

By rearranging the terms, we have

$$
\begin{aligned}
e^{h^*_{\mathrm{GLA}}(x,y)} &= \frac{(\mathsf{p}(y\mid x))^{\frac{1}{1-q}}}{(\mathsf{p}(y))^{\frac{1}{1-q}}}\frac{\sum_{y'\in\mathcal{Y}}e^{h^*_{\mathrm{GLA}}(x,y')+\frac{\log(\mathsf{p}(y'))}{1-q}}}{\sum_{y'\in\mathcal{Y}}(\mathsf{p}(y'\mid x))^{\frac{1}{1-q}}}\\
&= \left(\frac{\mathsf{p}(x\mid y)}{\mathsf{p}(x)}\right)^{\frac{1}{1-q}}\frac{\sum_{y'\in\mathcal{Y}}e^{h^*_{\mathrm{GLA}}(x,y')+\frac{\log(\mathsf{p}(y'))}{1-q}}}{\sum_{y'\in\mathcal{Y}}(\mathsf{p}(y'\mid x))^{\frac{1}{1-q}}}. && \text{(Bayes' theorem)}
\end{aligned}
$$

Thus, since the term $\frac{\sum_{y'\in\mathcal{Y}}e^{h^*_{\mathrm{GLA}}(x,y')+\frac{\log(\mathsf{p}(y'))}{1-q}}}{\sum_{y'\in\mathcal{Y}}(\mathsf{p}(y'\mid x))^{\frac{1}{1-q}}}$ does not depend on $y$, we obtain

$$
\mathsf{h}^*_{\mathrm{GLA}}(x) = \underset{y\in\mathcal{Y}}{\arg\max}\, h^*_{\mathrm{GLA}}(x,y) = \underset{y\in\mathcal{Y}}{\arg\max}\, e^{h^*_{\mathrm{GLA}}(x,y)} = \underset{y\in\mathcal{Y}}{\arg\max}\,\mathsf{p}(x\mid y).
$$

By Lemma 1, we know that the Bayes classifier $h^*_{\mathrm{bal}}$ of the Balanced loss satisfies that

$$
\mathsf{h}^*_{\mathrm{bal}} = \underset{y\in\mathcal{Y}}{\arg\max}\,\frac{\mathsf{p}(y\mid x)}{\mathsf{p}(y)} = \underset{y\in\mathcal{Y}}{\arg\max}\,\mathsf{p}(x\mid y).
$$

Therefore, we have $\mathsf{h}^*_{\mathrm{GLA}}(x)=\mathsf{h}^*_{\mathrm{bal}}$. This implies that the GLA loss $\ell_{\mathrm{GLA}}$ is Bayes-consistent with respect to the balanced loss $\ell_{\mathrm{BAL}}$. $\qquad\square$

# H  $\mathcal{H}$-Consistency for the GLA losses: proof of Theorem 4

**Theorem 4.** *Assume that $\mathcal{H}$ is complete. Then, for any $q \in [0, 1)$, the following $\mathcal{H}$-consistency bound holds for the GLA loss $\ell_{\mathrm{GLA}}$:*

$$\mathcal{R}_{\ell_{\mathrm{BAL}}}(h) - \mathcal{R}^*_{\ell_{\mathrm{BAL}}}(\mathcal{H}) + \mathcal{M}_{\ell_{\mathrm{BAL}}}(\mathcal{H}) \leq \Gamma\big(\mathcal{R}_{\ell_{\mathrm{GLA}}}(h) - \mathcal{R}^*_{\ell_{\mathrm{GLA}}}(\mathcal{H}) + \mathcal{M}_{\ell_{\mathrm{GLA}}}(\mathcal{H})\big),$$

*where $\Gamma(t) = \frac{\sqrt{2t}}{\mathsf{p}_{\min}}$ for $q = 0$, and $\Gamma(t) = \frac{\sqrt{2t}}{(\mathsf{p}_{\min})^{\frac{1}{1-q}}(1-q)^{\frac{1}{2}}}$ for $q \in (0, 1)$. In the special case where the approximation error $\mathcal{A}_{\ell_{\mathrm{GLA}}}(\mathcal{H}) = 0$, the bound simplifies to:*

$$\mathcal{R}_{\ell_{\mathrm{BAL}}}(h) - \mathcal{R}^*_{\ell_{\mathrm{BAL}}}(\mathcal{H}) \leq \Gamma\big(\mathcal{R}_{\ell_{\mathrm{GLA}}}(h) - \mathcal{R}^*_{\ell_{\mathrm{GLA}}}(\mathcal{H})\big),$$

*Proof.* The proof involves a characterization of the conditional regret of the balanced loss and the use of Gibbs distributions and Pinsker-type inequalities for analyzing GLA losses.

By Lemma 1, for complete hypothesis sets, the conditional regret of the balanced loss can be expressed as follows:

$$\Delta\mathcal{C}_{\ell_{\mathrm{BAL}},\mathcal{H}}(h, x) = \max_{y \in \mathcal{Y}} \frac{\mathsf{p}(y \mid x)}{\mathsf{p}(y)} - \frac{\mathsf{p}(\mathsf{h}(x)) \mid x)}{\mathsf{p}(\mathsf{h}(x))}.$$

Let $\mathsf{y}(x) = \operatorname{argmax}_{y \in \mathcal{Y}} \frac{\mathsf{p}(y|x)}{\mathsf{p}(y)}$, where we choose the label with the same deterministic strategy for breaking ties as that of $\mathsf{h}(x) = \operatorname{argmax}_{y \in \mathcal{Y}} h(x, y)$. We analyze by cases.

**Case I:** $q = 0$. In this case, the conditional regret for the GLA loss can be written as

$$\mathcal{C}_{\ell_{\mathrm{GLA}}}(h, x)) = -\sum_{y \in \mathcal{Y}} \mathsf{p}(y \mid x) \log\left(\frac{e^{h(x,y)+\log(\mathsf{p}(y))}}{\sum_{y' \in \mathcal{Y}} e^{h(x,y')+\log(\mathsf{p}(y'))}}\right) = -\sum_{y \in \mathcal{Y}} \mathsf{p}(y \mid x) \log\big(\overline{\mathcal{S}}(x, y)\big)$$

where we let $\overline{\mathcal{S}}(x, y) = \frac{e^{\overline{h}(x,y)}}{\sum_{y' \in \mathcal{Y}} e^{\overline{h}(x,y')}} \in [0, 1]$ for any $y \in \mathcal{Y}$ with $\overline{h}(x, y) = h(x, y) + \log(\mathsf{p}(y))$ and the constraint that $\sum_{y \in \mathcal{Y}} \overline{\mathcal{S}}(x, y) = 1$. Note that $\overline{\mathcal{S}}$ can be viewed as a Gibbs distribution induced by $h$ with prior $\mathsf{p}(y)$. Leveraging the facts that $\overline{\mathcal{S}}$ is a surjection and $\mathcal{H}$ is complete, minimizing over $\overline{\mathcal{S}}$, we know that $\mathcal{C}^*_{\ell_{\mathrm{GLA}}}(\mathcal{H}, x)$ has the following form:

$$\mathcal{C}^*_{\ell_{\mathrm{GLA}}}(\mathcal{H}, x) = -\sum_{y \in \mathcal{Y}} \mathsf{p}(y \mid x) \log(\mathsf{p}(y \mid x)).$$

Thus, we obtain

$$\begin{aligned}
\Delta\mathcal{C}_{\ell_{\mathrm{GLA}}\mathcal{H}}(h, x) &= \mathcal{C}_{\ell_{\mathrm{GLA}}}(h, x) - \mathcal{C}^*_{\ell_{\mathrm{GLA}}}(\mathcal{H}, x) \\
&= \sum_{y \in \mathcal{Y}} \mathsf{p}(y \mid x) \log(\mathsf{p}(y \mid x)) - \sum_{y \in \mathcal{Y}} \mathsf{p}(y \mid x) \log(\mathcal{S}(x, y)) \\
&= \sum_{y \in \mathcal{Y}} \mathsf{p}(y \mid x) \log(\mathsf{p}(y \mid x)) - \sum_{y \in \mathcal{Y}} \mathsf{p}(y \mid x) \log\left(\frac{e^{h(x,y)+\log(\mathsf{p}(y))}}{\sum_{y' \in \mathcal{Y}} e^{h(x,y')+\log(\mathsf{p}(y'))}}\right) \\
&= \sum_{y \in \mathcal{Y}} \mathsf{p}(y \mid x) \log\left(\mathsf{p}(y \mid x)\frac{\sum_{y' \in \mathcal{Y}} e^{h(x,y')+\log(\mathsf{p}(y'))}}{e^{h(x,y)+\log(\mathsf{p}(y))}}\right) \\
&= \mathsf{D}\big(\mathsf{p}(\cdot \mid x) \| \overline{S}(x, \cdot)\big)
\end{aligned}$$

where $\mathsf{D}(\mathsf{p}\|\mathsf{q})$ is the relative entropy of two distributions $\mathsf{p}$ and $\mathsf{q}$. Consider the case where $\mathsf{y}(x) \neq \mathsf{h}(x)$. Then, by Pinsker's inequality [Mohri et al., 2018, Proposition E.7], we have

$$\begin{aligned}
&\Delta\mathcal{C}_{\ell_{\mathrm{GLA}}\mathcal{H}}(h, x) \\
&= \mathsf{D}\big(\mathsf{p}(\cdot \mid x) \| \overline{S}(x, \cdot)\big) \\
&\geq \frac{1}{2}\big\|\mathsf{p}(\cdot \mid x) - \overline{S}(x, \cdot)\big\|_1^2 \qquad\qquad\qquad\qquad\qquad\qquad \text{(Pinsker's inequality)} \\
&\geq \frac{1}{2}\big(\big|\mathsf{p}(\mathsf{y}(x) \mid x) - \overline{S}(x, \mathsf{y}(x))\big| + \big|\mathsf{p}(\mathsf{h}(x) \mid x) - \overline{S}(x, \mathsf{h}(x))\big|\big)^2 \\
&= \frac{1}{2}\left(\mathsf{p}(\mathsf{y}(x))\left|\frac{\mathsf{p}(\mathsf{y}(x) \mid x)}{\mathsf{p}(\mathsf{y}(x))} - \frac{\overline{S}(x, \mathsf{y}(x))}{\mathsf{p}(\mathsf{y}(x))}\right| + \mathsf{p}(\mathsf{h}(x))\left|\frac{\mathsf{p}(\mathsf{h}(x) \mid x)}{\mathsf{p}(\mathsf{h}(x))} - \frac{\overline{S}(x, \mathsf{h}(x))}{\mathsf{p}(\mathsf{h}(x))}\right|\right)^2.
\end{aligned}$$

Plugging the expression of $\pi_h$, we have

$$\Delta\mathcal{C}_{\ell_{\mathrm{GLA}}\mathcal{H}}(h,x)$$

$$\geq \frac{1}{2}\left(\mathsf{p}(\mathsf{y}(x))\left|\frac{\mathsf{p}(\mathsf{y}(x)\mid x)}{\mathsf{p}(\mathsf{y}(x))}-\frac{e^{h(x,\mathsf{y}(x))}}{\sum_{y'\in\mathcal{Y}}e^{h(x,y')+\log(\mathsf{p}(y'))}}\right|+\mathsf{p}(\mathsf{h}(x))\left|\frac{\mathsf{p}(\mathsf{h}(x)\mid x)}{\mathsf{p}(\mathsf{h}(x))}-\frac{e^{h(x,\mathsf{h}(x))}}{\sum_{y'\in\mathcal{Y}}e^{h(x,y')+\log(\mathsf{p}(y'))}}\right|\right)^2$$

$$\geq \frac{(\mathsf{p}_{\min})^2}{2}\left(\left|\frac{\mathsf{p}(\mathsf{y}(x)\mid x)}{\mathsf{p}(\mathsf{y}(x))}-\frac{e^{h(x,\mathsf{y}(x))}}{\sum_{y'\in\mathcal{Y}}e^{h(x,y')+\log(\mathsf{p}(y'))}}\right|+\left|\frac{\mathsf{p}(\mathsf{h}(x)\mid x)}{\mathsf{p}(\mathsf{h}(x))}-\frac{e^{h(x,\mathsf{h}(x))}}{\sum_{y'\in\mathcal{Y}}e^{h(x,y')+\log(\mathsf{p}(y'))}}\right|\right)^2$$

$$\geq \frac{(\mathsf{p}_{\min})^2}{2}\left|\frac{\mathsf{p}(\mathsf{y}(x)\mid x)}{\mathsf{p}(\mathsf{y}(x))}-\frac{\mathsf{p}(\mathsf{h}(x)\mid x)}{\mathsf{p}(\mathsf{h}(x))}+\frac{e^{h(x,\mathsf{h}(x))}}{\sum_{y'\in\mathcal{Y}}e^{h(x,y')+\log(\mathsf{p}(y'))}}-\frac{e^{h(x,\mathsf{y}(x))}}{\sum_{y'\in\mathcal{Y}}e^{h(x,y')+\log(\mathsf{p}(y'))}}\right|^2$$

$$(|a|+|b|\geq|a-b|)$$

$$\geq \frac{(\mathsf{p}_{\min})^2}{2}\left|\frac{\mathsf{p}(\mathsf{y}(x)\mid x)}{\mathsf{p}(\mathsf{y}(x))}-\frac{\mathsf{p}(\mathsf{h}(x)\mid x)}{\mathsf{p}(\mathsf{h}(x))}\right|^2$$

$(\frac{\mathsf{p}(\mathsf{y}(x)|x)}{\mathsf{p}(\mathsf{y}(x))}-\frac{\mathsf{p}(\mathsf{h}(x)|x)}{\mathsf{p}(\mathsf{h}(x))}\geq 0$ and $\frac{e^{h(x,\mathsf{h}(x))}}{\sum_{y'\in\mathcal{Y}}e^{h(x,y')+\log(\mathsf{p}(y'))}}-\frac{e^{h(x,\mathsf{y}(x))}}{\sum_{y'\in\mathcal{Y}}e^{h(x,y')+\log(\mathsf{p}(y'))}}\geq 0$ by def. of $\mathsf{y}(x)$ and $\mathsf{h}(x)$)

$$= \frac{(\mathsf{p}_{\min})^2}{2}(\Delta\mathcal{C}_{\ell_{\mathrm{BAL}},\mathcal{H}}(h,x))^2.$$

Then, by taking the expectation on both sides and using the Jensen's inequality, we obtain

$$\mathcal{R}_{\ell_{\mathrm{BAL}}}(h)-\mathcal{R}^*_{\ell_{\mathrm{BAL}}}(\mathcal{H})+\mathcal{M}_{\ell_{\mathrm{BAL}}}(\mathcal{H})\leq\Gamma\big(\mathcal{R}_{\ell_{\mathrm{GLA}}}(h)-\mathcal{R}^*_{\ell_{\mathrm{GLA}}}(\mathcal{H})+\mathcal{M}_{\ell_{\mathrm{GLA}}}(\mathcal{H})\big),$$

where $\Gamma(t)=\frac{\sqrt{2t}}{\mathsf{p}_{\min}}$.

**Case II:** $q\in(0,1)$. In this case, the conditional regret for the GLA loss can be written as

$$\mathcal{C}_{\ell_{\mathrm{GLA}}}(h,x))=-\sum_{y\in\mathcal{Y}}\mathsf{p}(y\mid x)\Psi^q\left(\frac{e^{h(x,y)+\frac{\log(\mathsf{p}(y))}{1-q}}}{\sum_{y'\in\mathcal{Y}}e^{h(x,y')+\frac{\log(\mathsf{p}(y'))}{1-q}}}\right)=-\sum_{y\in\mathcal{Y}}\mathsf{p}(y\mid x)\Psi^q\big(\overline{\mathcal{S}}(x,y)\big)$$

where we let $\overline{\mathcal{S}}(x,y)=\frac{e^{\overline{h}(x,y)}}{\sum_{y'\in\mathcal{Y}}e^{\overline{h}(x,y')}}\in[0,1]$ for any $y\in\mathcal{Y}$ with $\overline{h}(x,y)=h(x,y)+\frac{\log(\mathsf{p}(y))}{1-q}$ and the constraint that $\sum_{y\in\mathcal{Y}}\overline{\mathcal{S}}(x,y)=1$. Note that $\overline{\mathcal{S}}$ can be viewed as a Gibbs distribution induced by $h$. Leveraging the facts that $\overline{\mathcal{S}}$ is a surjection and $\mathcal{H}$ is complete, minimizing over $\overline{\mathcal{S}}$, we know that $\mathcal{C}^*_{\ell_{\mathrm{GLA}}}(\mathcal{H},x)$ has the following form:

$$\mathcal{C}^*_{\ell_{\mathrm{GLA}}}(\mathcal{H},x)=\sum_{y\in\mathcal{Y}}\mathsf{p}(y\mid x)\Psi^q\left(\frac{\mathsf{p}(y\mid x)^{\frac{1}{1-q}}}{\sum_{y\in\mathcal{Y}}\mathsf{p}(y\mid x)^{\frac{1}{1-q}}}\right)=\frac{1}{q}\sum_{y\in\mathcal{Y}}\mathsf{p}(y\mid x)\left(1-\left(\frac{\mathsf{p}(y\mid x)^{\frac{1}{1-q}}}{\sum_{y\in\mathcal{Y}}\mathsf{p}(y\mid x)^{\frac{1}{1-q}}}\right)^q\right).$$

Thus, we obtain

$$\Delta\mathcal{C}_{\ell_{\mathrm{GLA}}\mathcal{H}}(h,x)$$

$$= \mathcal{C}_{\ell_{\mathrm{GLA}}}(h,x)-\mathcal{C}^*_{\ell_{\mathrm{GLA}}}(\mathcal{H},x)$$

$$= \frac{1}{q}\sum_{y\in\mathcal{Y}}\mathsf{p}(y\mid x)\big(1-\overline{S}(x,y)^q\big)-\frac{1}{q}\sum_{y\in\mathcal{Y}}\mathsf{p}(y\mid x)\left(1-\left(\frac{\mathsf{p}(y\mid x)^{\frac{1}{1-q}}}{\sum_{y\in\mathcal{Y}}\mathsf{p}(y\mid x)^{\frac{1}{1-q}}}\right)^q\right)$$

$$= \frac{\sum_{y\in\mathcal{Y}}\mathsf{p}(y\mid x)\left(\left(\frac{\mathsf{p}(y|x)^{\frac{1}{1-q}}}{\sum_{y\in\mathcal{Y}}\mathsf{p}(y|x)^{\frac{1}{1-q}}}\right)^q-\overline{S}(x,y)^q\right)}{q}$$

$$= \left(\sum_{y\in\mathcal{Y}}\mathsf{p}(y\mid x)^{\frac{1}{1-q}}\right)^{1-q}\frac{\left(1-\sum_{y\in\mathcal{Y}}\left(\frac{\mathsf{p}(y|x)^{\frac{1}{1-q}}}{\sum_{y\in\mathcal{Y}}\mathsf{p}(y|x)^{\frac{1}{1-q}}}\right)^{1-q}\overline{S}(x,y)^q\right)}{q}$$

$$= \left(\sum_{y\in\mathcal{Y}}\mathsf{p}(y\mid x)^{\frac{1}{1-q}}\right)^{1-q}\mathsf{T}_{1-q}\big(\mathsf{s}(\cdot\mid x)\|\overline{S}(x,\cdot)\big)$$

where $\mathsf{T}_q(\mathsf{p}\|\mathsf{q})$ denotes the Tsallis relative entropy of order $q$ between the distributions $\mathsf{p}$ and $\mathsf{q}$, and $\mathsf{s}(y \mid x) = \frac{\mathsf{p}(y|x)^{\frac{1}{1-q}}}{\sum_{y \in \mathcal{Y}} \mathsf{p}(y|x)^{\frac{1}{1-q}}}$. Consider the case where $\mathsf{y}(x) \neq \mathsf{h}(x)$. Then, by a Pinsker-type inequality [Rastegin, 2013, Eq. (4.13)], we have

$$\Delta \mathcal{C}_{\ell_{\mathrm{GLA}}\mathcal{H}}(h, x)$$

$$= \left(\sum_{y \in \mathcal{Y}} \mathsf{p}(y \mid x)^{\frac{1}{1-q}}\right)^{1-q} \mathsf{T}_{1-q}\big(\mathsf{s}(\cdot \mid x)\|\overline{S}(x, \cdot)\big)$$

$$\geq \frac{1-q}{2} \left(\sum_{y \in \mathcal{Y}} \mathsf{p}(y \mid x)^{\frac{1}{1-q}}\right)^{1-q} \big\|\mathsf{s}(\cdot \mid x) - \overline{S}(x, \cdot)\big\|_1^2$$

$$\text{(Pinsker-type inequality [Rastegin, 2013, Eq. (4.13)])}$$

$$\geq \frac{1-q}{2} \left(\sum_{y \in \mathcal{Y}} \mathsf{p}(y \mid x)^{\frac{1}{1-q}}\right)^{1-q} \big(\big|\mathsf{s}(\mathsf{y}(x) \mid x) - \overline{S}(x, \mathsf{y}(x))\big| + \big|\mathsf{s}(\mathsf{h}(x) \mid x) - \overline{S}(x, \mathsf{h}(x))\big|\big)^2$$

$$= \frac{1-q}{2} \left(\sum_{y \in \mathcal{Y}} \mathsf{p}(y \mid x)^{\frac{1}{1-q}}\right)^{1-q}$$

$$\times \left(\mathsf{p}(\mathsf{y}(x))^{\frac{1}{1-q}}\left|\frac{\mathsf{s}(\mathsf{y}(x) \mid x)}{\mathsf{p}(\mathsf{y}(x))^{\frac{1}{1-q}}} - \frac{\overline{S}(x, \mathsf{y}(x))}{\mathsf{p}(\mathsf{y}(x))^{\frac{1}{1-q}}}\right| + \mathsf{p}(\mathsf{h}(x))^{\frac{1}{1-q}}\left|\frac{\mathsf{s}(\mathsf{h}(x) \mid x)}{\mathsf{p}(\mathsf{h}(x))^{\frac{1}{1-q}}} - \frac{\overline{S}(x, \mathsf{h}(x))}{\mathsf{p}(\mathsf{h}(x))^{\frac{1}{1-q}}}\right|\right)^2$$

$$\geq \frac{1-q}{2}(\mathsf{p}_{\min})^{\frac{2}{1-q}} \left(\sum_{y \in \mathcal{Y}} \mathsf{p}(y \mid x)^{\frac{1}{1-q}}\right)^{1-q}$$

$$\times \left(\left|\frac{\mathsf{s}(\mathsf{y}(x) \mid x)}{\mathsf{p}(\mathsf{y}(x))^{\frac{1}{1-q}}} - \frac{\mathsf{s}(\mathsf{h}(x) \mid x)}{\mathsf{p}(\mathsf{h}(x))^{\frac{1}{1-q}}} + \frac{\overline{S}(x, \mathsf{h}(x))}{\mathsf{p}(\mathsf{h}(x))^{\frac{1}{1-q}}} - \frac{\overline{S}(x, \mathsf{y}(x))}{\mathsf{p}(\mathsf{y}(x))^{\frac{1}{1-q}}}\right|\right)^2 \qquad (|a| + |b| \geq |a - b|)$$

$$\geq \frac{1-q}{2}(\mathsf{p}_{\min})^{\frac{2}{1-q}} \left(\sum_{y \in \mathcal{Y}} \mathsf{p}(y \mid x)^{\frac{1}{1-q}}\right)^{1-q} \left|\frac{\mathsf{s}(\mathsf{y}(x) \mid x)}{\mathsf{p}(\mathsf{y}(x))^{\frac{1}{1-q}}} - \frac{\mathsf{s}(\mathsf{h}(x) \mid x)}{\mathsf{p}(\mathsf{h}(x))^{\frac{1}{1-q}}}\right|^2.$$

$$\left(\frac{\mathsf{s}(\mathsf{y}(x)|x)}{\mathsf{p}(\mathsf{y}(x))^{\frac{1}{1-q}}} - \frac{\mathsf{s}(\mathsf{h}(x)|x)}{\mathsf{p}(\mathsf{h}(x))^{\frac{1}{1-q}}} \geq 0 \text{ and } \frac{\overline{S}(x,\mathsf{h}(x))}{\mathsf{p}(\mathsf{h}(x))^{\frac{1}{1-q}}} - \frac{\overline{S}(x,\mathsf{y}(x))}{\mathsf{p}(\mathsf{y}(x))^{\frac{1}{1-q}}} \geq 0 \text{ by def. of } \mathsf{y}(x) \text{ and } \mathsf{h}(x)\right)$$

Plugging the expression of $\mathsf{s}(\cdot \mid x)$, we have

$$\Delta \mathcal{C}_{\ell_{\mathrm{GLA}}\mathcal{H}}(h, x)$$

$$\geq \frac{1-q}{2}(\mathsf{p}_{\min})^{\frac{2}{1-q}} \left(\sum_{y \in \mathcal{Y}} \mathsf{p}(y \mid x)^{\frac{1}{1-q}}\right)^{1-q} \left|\frac{\mathsf{s}(\mathsf{y}(x) \mid x)}{\mathsf{p}(\mathsf{y}(x))^{\frac{1}{1-q}}} - \frac{\mathsf{s}(\mathsf{h}(x) \mid x)}{\mathsf{p}(\mathsf{h}(x))^{\frac{1}{1-q}}}\right|^2$$

$$= \frac{1-q}{2}(\mathsf{p}_{\min})^{\frac{2}{1-q}} \left(\sum_{y \in \mathcal{Y}} \mathsf{p}(y \mid x)^{\frac{1}{1-q}}\right)^{-q-1} \left|\left(\frac{\mathsf{p}(\mathsf{y}(x) \mid x)}{\mathsf{p}(\mathsf{y}(x))}\right)^{\frac{1}{1-q}} - \left(\frac{\mathsf{p}(\mathsf{h}(x) \mid x)}{\mathsf{p}(\mathsf{h}(x))}\right)^{\frac{1}{1-q}}\right|^2$$

$$\leq \frac{1-q}{2}(\mathsf{p}_{\min})^{\frac{2}{1-q}} \left(\sum_{y \in \mathcal{Y}} \mathsf{p}(y \mid x)^{\frac{1}{1-q}}\right)^{-q-1} \left|\frac{\mathsf{p}(\mathsf{y}(x) \mid x)}{\mathsf{p}(\mathsf{y}(x))}\left(\frac{\mathsf{p}(\mathsf{y}(x) \mid x)}{\mathsf{p}(\mathsf{y}(x))}\right)^{\frac{q}{1-q}} - \frac{\mathsf{p}(\mathsf{h}(x) \mid x)}{\mathsf{p}(\mathsf{h}(x))}\left(\frac{\mathsf{p}(\mathsf{h}(x) \mid x)}{\mathsf{p}(\mathsf{h}(x))}\right)^{\frac{q}{1-q}}\right|^2$$

$$\geq \frac{1-q}{2}(\mathsf{p}_{\min})^{\frac{2}{1-q}} \left(\sum_{y \in \mathcal{Y}} \mathsf{p}(y \mid x)^{\frac{1}{1-q}}\right)^{-q-1} \left|\frac{\mathsf{p}(\mathsf{y}(x) \mid x)}{\mathsf{p}(\mathsf{y}(x))}\left(\frac{\mathsf{p}(\mathsf{y}(x) \mid x)}{\mathsf{p}(\mathsf{y}(x))}\right)^{\frac{q}{1-q}} - \frac{\mathsf{p}(\mathsf{h}(x) \mid x)}{\mathsf{p}(\mathsf{h}(x))}\left(\frac{\mathsf{p}(\mathsf{h}(x) \mid x)}{\mathsf{p}(\mathsf{h}(x))}\right)^{\frac{q}{1-q}}\right|^2$$

$$\geq \frac{1-q}{2}(\mathsf{p}_{\min})^{\frac{2}{1-q}} \left(\sum_{y \in \mathcal{Y}} \mathsf{p}(y \mid x)^{\frac{1}{1-q}}\right)^{-q-1} \left(\frac{\mathsf{p}(\mathsf{y}(x) \mid x)}{\mathsf{p}(\mathsf{y}(x))}\right)^{\frac{2q}{1-q}} \left|\frac{\mathsf{p}(\mathsf{y}(x) \mid x)}{\mathsf{p}(\mathsf{y}(x))} - \frac{\mathsf{p}(\mathsf{h}(x) \mid x)}{\mathsf{p}(\mathsf{h}(x))}\right|^2.$$

Next, using $\sum_{y\in\mathcal{Y}} \mathsf{p}(y\mid x)^{\frac{1}{1-q}} = \|p(\cdot\mid x)\|_{\frac{1}{1-q}}^{\frac{1}{1-q}} \le \|p(\cdot\mid x)\|_1^{\frac{1}{1-q}} = 1$ and $\frac{\mathsf{p}(\mathsf{y}(x)\mid x)}{\mathsf{p}(\mathsf{y}(x))} = \max_{y\in\mathcal{Y}} \frac{\mathsf{p}(y\mid x)}{\mathsf{p}(y)} \ge 1$,
we can write:

$$\Delta\mathcal{C}_{\ell_{\mathrm{GLA}}\mathcal{H}}(h,x) \ge \frac{1-q}{2}(\mathsf{p}_{\min})^{\frac{2}{1-q}}\left|\frac{\mathsf{p}(\mathsf{y}(x)\mid x)}{\mathsf{p}(\mathsf{y}(x))} - \frac{\mathsf{p}(\mathsf{h}(x)\mid x)}{\mathsf{p}(\mathsf{h}(x))}\right|^2$$

$$= \frac{1-q}{2}(\mathsf{p}_{\min})^{\frac{2}{1-q}}\left(\Delta\mathcal{C}_{\ell_{\mathrm{BAL}},\mathcal{H}}(h,x)\right)^2.$$

Then, by taking the expectation on both sides and using the Jensen's inequality, we obtain

$$\mathcal{R}_{\ell_{\mathrm{BAL}}}(h) - \mathcal{R}_{\ell_{\mathrm{BAL}}}^*(\mathcal{H}) + \mathcal{M}_{\ell_{\mathrm{BAL}}}(\mathcal{H}) \le \Gamma\left(\mathcal{R}_{\ell_{\mathrm{GLA}}}(h) - \mathcal{R}_{\ell_{\mathrm{GLA}}}^*(\mathcal{H}) + \mathcal{M}_{\ell_{\mathrm{GLA}}}(\mathcal{H})\right),$$

where $\Gamma(t) = \frac{\sqrt{2t}}{(\mathsf{p}_{\min})^{\frac{1}{1-q}}(1-q)^{\frac{1}{2}}}$. This concludes the first part of the proof. The second part follows directly using the fact that when the approximation error is zero: $\mathcal{A}_{\ell_{\mathrm{GLA}}}(\mathcal{H}) = 0$, the minimizability gap $\mathcal{M}_{\ell_{\mathrm{GLA}}}(\mathcal{H})$ vanishes. $\qquad\square$

# I  Margin bound: proof of Theorem 7

**Theorem 7** (Margin bound for cost-sensitive classification). *Let $\mathcal{H}$ be a family of functions mapping from $\mathcal{X}\times[n]$ to $\mathbb{R}$. Then, for any $\delta > 0$, with probability at least $1-\delta$, each of the following inequalities holds for all $h \in \mathcal{H}$:*

$$\mathcal{R}_{\mathsf{L}}(h) \le \widehat{\mathcal{R}}_{S,\rho}(h) + 4\overline{C}\sqrt{2n}\,\mathfrak{R}_m(\mathcal{H}) + \sqrt{\frac{\log\frac{1}{\delta}}{2m}}$$

$$\mathcal{R}_{\mathsf{L}}(h) \le \widehat{\mathcal{R}}_{S,\rho}(h) + 4\overline{C}\sqrt{2n}\,\widehat{\mathfrak{R}}_S(\mathcal{H}) + 3\sqrt{\frac{\log\frac{2}{\delta}}{2m}}.$$

*Proof.* Consider the family of functions taking values in $[0,1]$:

$$\mathcal{H}' = \{z = (x,y) \mapsto \mathsf{L}_\rho(h,x,y)\colon h \in \mathcal{H}\}.$$

By [Mohri et al., 2018, Theorem 3.3], with probability at least $1 - \delta$, for all $g \in \mathcal{H}'$,

$$\mathbb{E}[g(z)] \le \frac{1}{m}\sum_{i=1}^m g(z_i) + 2\widehat{\mathfrak{R}}_S(\mathcal{H}') + 3\sqrt{\frac{\log\frac{2}{\delta}}{2m}},$$

and thus, for all $h \in \mathcal{H}$,

$$\mathbb{E}[\mathsf{L}_\rho(h,x,y)] \le \widehat{\mathcal{R}}_{S,\rho}(h) + 2\widehat{\mathfrak{R}}_S(\mathcal{H}') + 3\sqrt{\frac{\log\frac{2}{\delta}}{2m}}.$$

Since $\mathcal{R}_{\mathsf{L}}(h) \le \mathcal{R}_{\mathsf{L}_\rho}(h) = \mathbb{E}[\mathsf{L}_\rho(h,x,y)]$, we have

$$\mathcal{R}_{\mathsf{L}}(h) \le \widehat{\mathcal{R}}_{S,\rho}(h) + 2\widehat{\mathfrak{R}}_S(\mathcal{H}') + 3\sqrt{\frac{\log\frac{2}{\delta}}{2m}}.$$

Fix $h$, $(x_i,y_i)$ and $\rho > 0$, define $\Psi$ as follows:

$$\Psi([h(x_i,y)]_{y\in[n]}) = c(x_i,y_i)\max_{y'\in[n]}\{\Phi_\rho(h(x_i,y_i) - h(x_i,y'))\}.$$

Then, by the sub-additivity of the maximum operator, we can write for any $f, \widetilde{f} \in \mathcal{H}$:

$$\Psi([h(x_i,y)]_{y\in[n]}) - \Psi([\widetilde{h}(x_i,y)]_{y\in[n]})$$

$$\le c(x_i,y_i)\max_{y'\in[n]}\{\Phi_\rho(h(x_i,y_i) - h(x_i,y'))\} - c(x_i,y_i)\max_{y'\in[n]}\{\Phi_\rho(\widetilde{h}(x_i,y_i) - \widetilde{h}(x_i,y'))\}$$

$$\le \frac{2c(x_i,y_i)}{\rho}\left\{\left\|[h(x_i,y) - \widetilde{h}(x_i,y)]_{y\in[n]}\right\|_1\right\} \qquad\qquad (\text{by }\tfrac{1}{\rho}\text{-Lipschitzness of } \Phi_\rho)$$

$$\le \frac{2\overline{C}\sqrt{n}}{\rho}\left\|[h(x_i,y) - \widetilde{h}(x_i,y)]_{y\in[n]}\right\|_2.$$

Thus, $\Psi$ is $\frac{2\sqrt{n}}{\rho}$-Lipschitz with respect to the $\|\cdot\|_2$ norm. Thus, by the vector contraction lemma [Maurer, 2016, Cortes et al., 2016], $\widehat{\mathfrak{R}}_S(\mathcal{H}')$ can be bounded as follows:

$$\widehat{\mathfrak{R}}_S(\mathcal{H}') \leq 2\overline{C}\sqrt{2n}\,\widehat{\mathfrak{R}}_S(\mathcal{H}).$$

This proves the second inequality. The first inequality, can be derived in the same way by using the first inequality of [Mohri et al., 2018, Theorem 3.3]. $\qquad\square$

