# OpenReview forum: "Improved Balanced Classification with Theoretically Grounded Loss Functions"
_NeurIPS.cc/2025/Conference — NeurIPS 2025 poster_

### Official Review · Reviewer_UxSp · 2025-06-13

**Clarity:** 2
**Significance:** 4
**Originality:** 4
**Rating:** 5
**Confidence:** 3

**Summary:**

This paper motivates a new algorithm for conducting class-balanced cross entropy (CE) minimization. The paper states that class-balanced CE minimization has been addressed via heuristic methods such as class-weighted CE and logit-adjusted loss, but that these methods lack desired theoretical properties such as Bayes consistency. The paper then develops two new loss functions - Generalized Logit-Adjusted Loss (GLA) and Generalized Class-Aware Loss (GCA), as algorithms that upper bound the true cost-sensitive loss while having strong theoretical properties of Bayes consistency and H-consistency. The experimental results show that the new loss functions perform the best in terms of balanced-error when used in multi-class classification, for different kinds of class-skew in the training data.

**Questions:**

The paper is as a whole excellent and significant, the following are my suggestions to make it more accessible to readers not as familiar with this style of work and further increase adoption:
1. Add further details for why class-weighted CE does not work well. This is (in my experience) by far the most common approach to handling this very general problem. Identifying the errors more clearly can go a long way to further boost the significance of the results. For example, the graph on the (lack of) H-consistency of LA loss is great.
2. When introducing the GCA loss, it says that $\rho$ is "is a vector of positive confidence margin parameters for each class." Practically speaking, what is this supposed to be? More hyperparameters typically yields more complications when trying to use a loss in practice, so helping clarify what these are meant to represent may provide more transparency around using this loss.
3. The proofs could really use more annotation. A "plan" before the proof starts would go a long way, as would more details about how one step flows into the next. There appear to be some interesting techniques used, such as in the H-consistency results of GCA where the authors transform GCA to GCE via the change of measure and then use known consistency results. Hence, it would be really helpful to have some description.
4. Most of the notation seems to be clearly defined, but I could not find where $\mathcal{E}$ was defined. I'm guessing it's the empirical loss, but overall I recommend one more parse through the paper to check that notation is clearly defined.

**Ethical Concerns:**

["NO or VERY MINOR ethics concerns only"]

**Final Justification:**

This paper overall presents a novel solution to a fairly foundational problem in ML - weighted class cross entropy. The solution and techniques used in the paper are interesting to read and provide a theoretically justified and practical approach to improving solving class imbalance classification problems. My main concerns were around the accessibility and clarity of the paper, but provided that the authors work to improve the interpretability and more hand-holding through the relatively dense proofs, my rating is an accept.

**Limitations:**

Yes

**Quality:**

4

**Strengths And Weaknesses:**

**Quality**: This paper is overall excellent. The GLA and GCA algorithms are very well-grounded, as they are motivated by the true balanced loss and naturally derived as being tractable surrogates through manipulation of the margin loss. Additionally, the theorems go beyond traditional convergence under convexity style proofs and show Bayes consistency/H-consistency, adding another layer of theoretical backing to the algorithms. The experiments are well-conducted and benchmarks against several well-known, relevant algorithms in this domain (e.g. focal loss, logit-adjusted, class-weighted CE, etc.). The experimental setting is slightly limited (three CV datasets) but is nonetheless appropriate and well-justified, as there are indeed several classes and the authors clearly explain how class imbalanced is induced. The proofs are difficult to read as they do not have much annotation, but from what I can gauge, they appear correct and have a logical development (e.g. $\rho$-margin loss -> upper bound -> GLA loss). The differences between the GLA and GCA loss are also clearly elucidated.

**Clarity**: Despite the excellent theoretical results, this paper was difficult to parse in multiple parts and needs some improvement (although I fully acknowledge the space limitations). At a high level, the lack of clarity is in some details, such as why weighted CE does not work well when data is easily separable, what the confidence margin parameters are supposed to represent, what the evaluation "balanced error" actually is, etc. Most importantly the proofs are hard to read. More details in the suggestions.

**Significance**: This is a very significant advancement in the development of class-balanced error minimization. Class-balanced error indeed has several applications, including in fairness, fraud-detection, rare-event detection, etc. In practice, class-weighted CE is used (or equivalently, data resampling), but this paper proposes a new path forward that presents a more theoretically disciplined, yet tractable approach. The proofs are also quite novel, and may be useful for future research in this area.

**Originality**: This paper is very original. It appropriately reviews existing literature around this topic and identifies the key works. The methods use to develop the GCA/GLA are relatively novel, as they take a principled approach to upper bounding the class-weighted 0/1 loss and develop a surrogate accordingly. I am not as familiar with proofs in Bayes consistency/H-consistency to judge if they are novel, but considering these results are rarely developed in research regarding loss-functions, they feel novel to me.

---

> ### Author Rebuttal · Authors · 2025-07-30
>
> Thank you for your appreciation of our work. We will take your suggestions into account when preparing the final version. Below please find responses to specific questions.
>
> **Strengths And Weaknesses: Quality ... Originality.**
>
> **Response:** We sincerely thank the reviewer for their positive assessment of the paper’s quality, significance, and originality. We appreciate your recognition of the principled derivation of GLA and GCA, the theoretical contributions on Bayes and H-consistency, and the relevance of our empirical evaluation.
>
> Regarding clarity, we agree that certain parts of the paper would benefit from improved exposition. In the final version, we will use the additional page to enhance the presentation for readers. Specifically, we will clarify why weighted cross-entropy can perform poorly in the separable regime, explain the interpretation of the confidence margin parameters (see responses to specific questions for more details), define and motivate the “balanced error” metric (i.e., the average of the balanced loss over the test data), and provide more intuitive explanations to help guide the reader through the proofs. Thank you again for your constructive feedback.
>
> **Questions:**
>
> **1. Add further details for why class-weighted CE does not work well. This is (in my experience) by far the most common approach to handling this very general problem. Identifying the errors more clearly can go a long way to further boost the significance of the results. For example, the graph on the (lack of) H-consistency of LA loss is great.**
>
> **Response:** Thank you for the insightful comment. In separable cases, class-weighted cross-entropy may still yield solutions with zero training loss that do not adjust decision boundaries meaningfully toward minority or majority classes. This is because class weighting does not influence the classifier once perfect separation is achieved. As a result, the method fails to address imbalance in such regimes. We will add further details and discussion of this issue in the final version to better highlight the limitations of class-weighted CE.
>
> **2. When introducing the GCA loss, it says that $\rho$ is "a vector of positive confidence margin parameters for each class." Practically speaking, what is this supposed to be? More hyperparameters typically yield more complications when trying to use a loss in practice, so helping clarify what these are meant to represent may provide more transparency around using this loss.**
>
> **Response:** Thank you for the thoughtful question. The confidence margin parameters $\rho_y$ allow for fine-grained adjustment of decision boundaries on a per-class basis. Specifically, by scaling the logit differences $h(x, y) − h(x, y')$ by $\rho_y$, the GCA loss effectively adjusts margins in a class-specific way. This transformation enables the model to better distinguish between dominant and rare classes, as it modulates how confidently each class needs to be separated. As highlighted by Cortes et al. (2025), such margin adjustments play a key role in shifting decision boundaries and mitigating imbalance, addressing limitations of simpler class-weighting schemes.
>
> We followed (Cortes et al., 2025) in using $\rho_k = m_k^{1/3}$ in our experiments. A similar derivation to theirs can be adapted to our setting, showing these values are theoretically optimal in the separable case, thereby providing a principled default. Empirically, GCA losses are robust to variations in $\rho_k$ around these values, though they can be tuned via cross-validation if desired. We will further elaborate on that in the final version.
>
> **3. The proofs could really use more annotation. A "plan" before the proof starts would go a long way, as would more details about how one step flows into the next. There appear to be some interesting techniques used, such as in the H-consistency results of GCA where the authors transform GCA to GCE via the change of measure and then use known consistency results. Hence, it would be really helpful to have some description.**
>
> **Response:** Thank you for the helpful suggestion. We agree that adding more annotations and an outline before key proofs would significantly improve readability. In the final version, we will provide a brief “proof plan” before each major result to explain the high-level strategy and highlight key steps and techniques, such as the change of measure used to relate GCA to GCE in the $H$-consistency proof. We will also add more intermediate explanations to clarify how each step follows, especially in technically involved arguments.
>
> **4. Most of the notation seems to be clearly defined, but I could not find where $\mathcal{E}$ was defined. I'm guessing it's the empirical loss, but overall I recommend one more parse through the paper to check that notation is clearly defined.**
>
> **Response:** Yes, $\widetilde{\mathcal{E}}$ refers to the empirical loss. We will do another careful pass through the manuscript to ensure all notation is clearly defined and consistently used. Thank you for bringing this to our attention.

---

> ### Comment · Reviewer_UxSp · 2025-08-08
> **Reviewer response**
>
> Thank you authors for the thorough response! It's great to hear that the authors will focus on improving clarity for the final version, as there are several very interesting things presented in this work. Making it more accessible will hopefully allow greater adoption of this work, as the core problem this paper is addressing is quite fundamental and useful in a multitude of cases.

---

### Official Review · Reviewer_S8CM · 2025-06-30

**Clarity:** 4
**Significance:** 3
**Originality:** 3
**Rating:** 5
**Confidence:** 2

**Summary:**

This paper proposes two novel surrogate loss functions for multi-class classification under class imbalance: Generalized Logit-Adjusted (GLA) loss and Generalized Class-Aware weighted (GCA) loss, building on the existing general cross-entropy (GCE) framework (Mao et al. ICML 2023). GLA losses generalize Logit-Adjusted losses, which shift logits based on class priors, to the broader general cross-entropy loss family. It includes theoretical analysis of consistency of the two proposed surrogate loss functions. Extensive experiments are conducted to evaluate the performance of GLA and GCA, in comparison with existing surrogate loss functions, demonstrating different operating characteristics of GLA and GCA.

**Questions:**

1) How to assess whether the key assumptions underlying the theoretical results in Section 5 hold in real-world data? Under what real-world settings are these assumptions plausible?

2) How do the proposed GCE and GCA compared to IMMAX (Cortes et al. 2025) in terms of their theoretical properties?

3) It would strengthen the experiemental results to more recent baseline methods such as
IMMAX (Cortes et al. 2025) and LSC (Wei et al., 2024).

**Ethical Concerns:**

["NO or VERY MINOR ethics concerns only"]

**Final Justification:**

I don't have additional major comments and will maintain my positive score for this paper.

**Limitations:**

yes

**Quality:**

4

**Strengths And Weaknesses:**

Strengths: Building on the existing general cross-entropy (GCE) framework (Mao et al. ICML 2023), this work proposes and studies two surrogate loss functions, the generalized logit-adjusted (GLA) loss and the generalized class-aware (GCA) loss. In GCA, $\rho$ is motivated by Cortes et al. (2025). Strong theoretical results are provided for consistency of GLA and GCA, showing that GLA is Bayes-consistent, but only $\mathcal{H}$-consistent for unbounded and complete hypothesis sets whereas GCA is $\mathcatl{H}-consistent for any hypothesis set that is bounded or complete and offers significantly stronger theoretical guarantees in imbalanced settings. Extensive experiments are conducted to demonstrate the superior performance of GLA and GCA in comparison with existing surrogate loss functions and also show different operating characteristics of GLA and GCA.


Weakness: It is unclear why directly optimizing the balanced classification loss is intractable.

It is unclear whether the key assumptions underlying the theoretical results in Section 5 are plausible in real-world data.

It is unclear How the proposed GCE and GCA compared to IMMAX (Cortes et al. 2025) and other existing surrogate losses in terms of theoretical properties.

Section 6: The numerical experiments did not include more recent baseline methods such as  IMMAX (Cortes et al. 2025) and LSC (Wei et al., 2024).

---

> ### Author Rebuttal · Authors · 2025-07-30
>
> Thank you for your appreciation of our work. We will take your suggestions into account when preparing the final version. Below please find responses to specific questions.
>
> **1. Weakness 1. It is unclear why directly optimizing the balanced classification loss is intractable.**
>
> **Response:** Directly optimizing the balanced classification loss is intractable because it involves an indicator function over the prediction and true label, making the objective discrete and non-differentiable. This is analogous to the standard 0-1 loss, which is also known to be intractable to optimize directly in the literature.
>
> **2. Weakness 2. It is unclear whether the key assumptions underlying the theoretical results in Section 5 are plausible in real-world data.**
>
> **Question 1. How to assess whether the key assumptions underlying the theoretical results in Section 5 hold in real-world data? Under what real-world settings are these assumptions plausible?**
>
> **Response:** The assumptions in Section 5 primarily concern properties of the hypothesis set. These are standard and typically satisfied in practice. Most natural hypothesis sets, such as linear models, neural networks, and the set of all measurable functions, are regular, meaning they produce predictions across all $n$ classes. Whether a hypothesis set is bounded or complete depends on the modeling choice (e.g., bounded weights in linear models). Importantly, our results do not assume any specific data distribution and hold for arbitrary distributions, including those arising in real-world settings.
>
> **3. Weakness 3. It is unclear how the proposed GCE and GCA compared to IMMAX (Cortes et al. 2025) and other existing surrogate losses in terms of theoretical properties.**
>
> **Question 2. How do the proposed GCE and GCA compared to IMMAX (Cortes et al. 2025) in terms of their theoretical properties?**
>
> **Response:** The key difference is that IMMAX (Cortes et al., 2025) is designed for optimizing the standard multi-class 0-1 loss under imbalanced data, whereas the proposed GCE and GCA losses are designed to optimize the balanced loss. As a result, IMMAX enjoys consistency with respect to the standard 0-1 loss, while GCE and GCA are consistent with respect to the balanced loss, a property most existing surrogate losses lack, as discussed in Section 3.3.
>
> **4. Weakness 4. Section 6: The numerical experiments did not include more recent baseline methods such as IMMAX (Cortes et al. 2025) and LSC (Wei et al., 2024).**
>
> **Question 3. It would strengthen the experimental results to more recent baseline methods such as IMMAX (Cortes et al. 2025) and LSC (Wei et al., 2024).**
>
> **Response:** IMMAX is designed to optimize the standard multi-class 0-1 loss under imbalanced data, whereas our proposed losses are specifically tailored to optimize the balanced loss. Therefore, we expect IMMAX to be inferior to our methods in terms of balanced error, which is our primary objective. We are happy to include comparisons with IMMAX (Cortes et al. 2025) and LSC (Wei et al., 2024) in the final version.

---

> > ### Comment · Reviewer_S8CM · 2025-08-05
> >
> > I want to thank the authors for addressing my comments.

---

### Official Review · Reviewer_Swme · 2025-07-02

**Clarity:** 4
**Significance:** 2
**Originality:** 2
**Rating:** 5
**Confidence:** 3

**Summary:**

The authors present generalized versions of the logit-adjust (LA) loss and the class-aware (CA) loss. They extend these through the generalized cross-entropy framework and show that they admit strong theoretical guarantees in terms of bayes consistency and $\mathcal{H}$ consistency.  The authors also demonstrate the efficacy of their methods on CIFAR10 and 100 with imbalance.

**Questions:**

1. How well-behaved are these loss functions in terms of optimization? Similar methods like $\alpha$-loss (Sypherd et al.) can make the optimization procedure quite slow in practice.
2. Ablations demonstrating the impact of the choice of $q$ could help readers better internalize its meaning and importance for tuning.
3. Ablations for the inclusion of $\rho$ in GCA would also be helpful.
4. A key shortcoming of standard weighted cross entropy is in the highly overparameterized setting. How do GCA and GLA fare when data is limited?
5. The weighting factor for GCA need not be the class prior. Tuning over this would be very interesting and give insights into whether the prior is an optimal choice for $q\ne1$.

**Ethical Concerns:**

["NO or VERY MINOR ethics concerns only"]

**Final Justification:**

Additional discussion of the proof novelty will be included in the revision which will greatly strengthen the results. Additionally, the authors' rebuttal addressed my major concerns and those of other reviewers. I believe this work will be interesting to the community at large.

**Limitations:**

yes

**Quality:**

3

**Strengths And Weaknesses:**

# Strengths
1. The paper is clear and well-written
2. The theoretical guarantees for both GLA and GCA are quite strong, and the proofs themselves have novelty. The favorable scaling of GCA with $p_\text{min}$ is especially compelling.
3. The empirical performance of these methods is compelling, especially compared with more specialized methods like focal loss.

# Weaknesses
1. The novelty of the generalization is somewhat limited, simply be a simple extension of LA and CA under the GCE framework.
2. The additional positive margins $\rho$ included in GCA are new hyperparameters to tune, and their impact is not completely clear. Similarly with $q$ in both GLA and GCA. Including ablations to demonstrate the sensitivity to these choices would be helpful.
3. The choice of the inverse of the prior for GCA is not clearly justified in the setting of $q\ne1$ as it could be for $q=1$.

---

> ### Author Rebuttal · Authors · 2025-07-30
>
> Thank you for your appreciation of our work. We will take your suggestions into account when preparing the final version. Below please find responses to specific questions.
>
> **Weaknesses:**
>
> **1. The novelty of the generalization is somewhat limited, simply be a simple extension of LA and CA under the GCE framework.**
>
> **Response:** The novelty of the generalization lies in ensuring that the extended formulations satisfy consistency guarantees. For example, the $(1 - q)$ factor in Eq. (4) is essential to the definition of GLA losses, ensuring Bayes-consistency for any $q \in [0, 1)$ (as shown in Section 5.2). This offers greater flexibility than the original LA loss, which is only Bayes-consistent when $\tau = 1$. Moreover, GLA losses enjoy stronger $H$-consistency guarantees when the hypothesis set $H$ is complete. We also refer to our response to Reviewer oVh8 (“Weakness 1”) for a more detailed discussion of the novel proof techniques used to establish these guarantees.
>
> **2. The additional positive margins $\rho$ included in GCA are new hyperparameters to tune, and their impact is not completely clear. Similarly with $q$ in both GLA and GCA. Including ablations to demonstrate the sensitivity to these choices would be helpful.**
>
> **Question 2. Ablations demonstrating the impact of the choice of $q$ could help readers better internalize its meaning and importance for tuning.**
>
> **Question 3. Ablations for the inclusion of $\rho$ in GCA would also be helpful.**
>
> **Response:** We followed (Cortes et al., 2025) in using $\rho_k = m_k^{1/3}$ in our experiments. A similar derivation to theirs can be adapted to our setting, showing these values are theoretically optimal in the separable case, thereby providing a principled default. Empirically, GCA losses are robust to variations in $\rho_k$ around these values, though they can be tuned via cross-validation if desired. We will further elaborate on that in the final version.
>
> For the parameter $q$ in both GLA and GCA, we selected values from ${0.1, 0.2, \ldots, 0.9}$, which are standard choices within the general cross-entropy family. Its performance depends on dataset imbalance (e.g., long-tailed vs. step imbalance).
> We will clarify these points and include additional ablation analysis in the final version.
>
> **3. The choice of the inverse of the prior for GCA is not clearly justified in the setting of $q \neq 1$ as it could be for $q = 1$.**
>
> **Response:** The motivation for using the inverse of the prior in GCA remains the same for $q \neq 1$ as for $q = 1$. The parameter $q$ simply specifies a particular loss within the generalized cross-entropy family, applicable in both standard and imbalanced settings. The inverse of the prior is used to align with the definition of the balanced loss, which reduces the influence of class imbalance by reweighting each example's error accordingly. This ensures that GCA losses benefit from consistency guarantees with respect to the balanced loss. We will clarify this in the final version.
>
> **Questions:**
>
> **1. How well-behaved are these loss functions in terms of optimization? Similar methods like $\alpha$-loss (Sypherd et al.) can make the optimization procedure quite slow in practice.**
>
> **Response:** Our loss functions are adapted from the general cross-entropy family and share similar convergence behavior when optimized with standard methods such as SGD, Adam, and AdaGrad. We will clarify this and provide more discussion in the final version.
>
> **4. A key shortcoming of standard weighted cross entropy is in the highly overparameterized setting. How do GCA and GLA fare when data is limited?**
>
> **Response:** Our study did not aim to specifically favor either overparameterized or standard parameterized settings. Rather, we closely followed the experimental setup used in prior work cited in Section 6 to ensure comparability. Investigating the behavior of GCA and GLA in overparameterized or data-limited regimes is indeed a compelling direction for future work. Regarding weighted cross-entropy, we note that such methods have been shown empirically not to be effective in general, including in non-overparameterized settings, depending on various factors (e.g., Van Hulse et al., 2007).
>
> **5. The weighting factor for GCA need not be the class prior. Tuning over this would be very interesting and give insights into whether the prior is an optimal choice for $q \neq 1$.**
>
> **Response:** From a theoretical perspective, the use of the inverse of the prior in GCA for $q \neq 1$ is motivated by the same reasoning as for $q = 1$: it aligns with the definition of the balanced loss and ensures consistency guarantees. However, from a practical standpoint, exploring alternative weighting schemes is indeed interesting and could provide insight into whether the prior is optimal. We will consider including this analysis in the final version.

---

> > ### Comment · Reviewer_Swme · 2025-08-04
> >
> > Thank you for the thorough consideration of the review. Additional discussion of the novelty of the proof technique would greatly strengthen the work and I am confident that the authors will add this information to the revision. My remaining concerns are mostly focused on the choice of weighting with GCA, which I believe could be explored in the revision following point 5 above. It seems intuitive to me that difference $q$ choices could lead to different weighting tradeoffs. Overall, the authors have addressed my concerns and I am raising my score to 5 accordingly.

---

> ### Author Response · Authors · 2025-08-04
> **Thank You**
>
> Thank you for your thoughtful feedback and for raising your score. We will include a clearer discussion of the novelty of our proof technique and expand on the tradeoffs in weighting choices for GCA as suggested.

---

### Official Review · Reviewer_oVh8 · 2025-07-03

**Clarity:** 3
**Significance:** 2
**Originality:** 2
**Rating:** 4
**Confidence:** 3

**Summary:**

The authors give two novel surrogate loss function families for addressing class imbalance in multi-class classification: Generalized Logit-Adjusted (GLA) losses and Generalized Class-Aware weighted (GCA) losses.

The authors provide theoretical analysis showing that GLA losses are Bayes-consistent but only $H$-consistent for complete hypothesis sets with bounds scaling as $1/p_{min}$, while GCA losses are $H$-consistent for both bounded and complete hypothesis sets with more favorable bounds scaling as $1/\sqrt{p_{min}}$.

Experimental results on image datasets show that both loss families outperform existing baselines.

**Questions:**

Line 14: "H-consistent for unbounded and complete hypothesis sets" - The conjunction "and" is confusing here. Should clarify if both conditions are required or if it's "unbounded or complete."

Line 92: "The margin $\rho_h(x, y)$ for a predictor $h \in H$" - The margin notation $\rho$ conflicts with the imbalance ratio $\rho$ used later.  Could you clarify?

Line 130: The LA loss definition uses $\tau$ but the text doesn't clearly explain what happens when $\tau \neq 1$ until much later.

Line 177: $\rho = (\rho_1, \ldots, \rho_n)$ - This reuses the margin notation from Page 3.

Line 191: "Note that while the $\rho_k$ values can be freely tuned" - What do you mean?

Line 258: The bound $\Gamma(t) = \sqrt{\frac{2t}{p_{min}}}$ -  Is this a tight bound?

Line 283: The theorem statement is quite dense and could benefit from more intuitive explanation.

Line 343: "imbalance ratio, $\rho = \frac{\max_{k=1}^c m_k}{\min_{k=1}^c m_k}$" - The notation $c$ for number of classes conflicts with the usage of $n$ elsewhere.

Line 351: "200 epochs" - No justification for this choice or sensitivity analysis.

Eq (5): The confidence margin parameters $\rho_y$ should be clearly distinguished from the margin function $\rho_h(x,y)$ defined earlier.

How sensitive are the GCA losses to the choice of confidence margin parameters $\rho_y$?

What is the computational complexity comparison between GLA and GCA losses?

Are there convergence guarantees for the optimization of these loss functions?

It would be great if the authors clarify the proof novelty compared to Mohri et al., 2018 and Mao et al., 2023b,a.

**Ethical Concerns:**

["NO or VERY MINOR ethics concerns only"]

**Final Justification:**

The authors have addressed most of my concerns. Some concerns remain regarding the novelty of the proofs with respect to Mohri et al. (2018) and Mao et al. (2023a, 2023b)

**Limitations:**

N.A.

**Paper Formatting Concerns:**

N.A.

**Quality:**

3

**Strengths And Weaknesses:**

Strengths:

The paper provides rigorous theoretical analysis (in the supplement) with well-established consistency results.

The presentation is good, especially Sections 2 and 3, which provide a good foundation for the main results of the paper.

The experiments include proper statistical reporting.  They compare against relevant baselines across different datasets.

Weaknesses:

The theoretical novelty is limited. It would be great if the authors clarify the proof novelty compared to Mohri et al., 2018 and Mao et al., 2023b,a in the main part of the paper.

The experiments are restricted to image classification tasks (CIFAR-10/100, Tiny ImageNet) with relatively simple architectures (ResNet-32). The generalizability to other domains and modern architectures is unclear.

The paper mentions that $\rho_k$ values in GCA losses can be tuned via cross-validation but provides limited guidance on computational complexity or sensitivity analysis. The suggestion to use $[m_k^{1/3}]_k$ as starting points without any intuitions theoretical justification.


While the paper mentions training details, it doesn't thoroughly analyze the computational cost of the proposed methods compared to baselines.

---

> ### Author Rebuttal · Authors · 2025-07-30
>
> Thank you for your appreciation of our work. We will take your suggestions into account when preparing the final version. Below please find responses to specific questions.
>
> **Weaknesses:**
>
> **1. The theoretical novelty is limited. It would be great if the authors clarify the proof novelty compared to Mohri et al., 2018 and Mao et al., 2023b,a in the main part of the paper.**
>
> **Response:** Mohri et al. (2018) presented margin bounds for standard multi-class classification. In contrast, we derive new margin bounds for cost-sensitive classification, a setting that introduces additional complexity due to the presence of instance-dependent cost functions. This requires the development of new proof techniques, including the derivation of an upper bound on the loss function expressed in terms of a margin loss and a maximum operator, along with an analysis of the Rademacher complexity of this maximum term via the vector contraction lemma. Moreover, in addition to the resulting margin bounds for GCA loss functions, our margin bounds for GLA loss functions are non-trivial and require a specific and entirely new analysis (Appendix B.2).
>
> Mao et al. (2023a,b) studied H-consistency bounds for loss functions in the general cross-entropy (GCE) family with respect to the standard multi-class 0-1 loss. In contrast, our work establishes H-consistency bounds for the proposed GCA and GLA losses with respect to the balanced loss, where both the surrogate and target losses are more complex. This required several novel technical contributions, including a characterization of the conditional regret of the balanced loss, the use of Gibbs distributions and Pinsker-type inequalities for analyzing GLA losses, and a reduction of the conditional regrets of the balanced and GCA losses to those of the 0-1 and GCE losses under a newly defined distribution.
>
> We will further elaborate on the novelty of our proof techniques and provide a more detailed account of our technical contributions in the main body of the final version, making use of the additional page limit.
>
> **2. The experiments are restricted to image classification tasks (CIFAR-10/100, Tiny ImageNet) with relatively simple architectures (ResNet-32). The generalizability to other domains and modern architectures is unclear.**
>
> **Response:** We followed closely the experimental setup of prior work cited in Section 6, using standard image benchmarks (CIFAR-10/100, Tiny ImageNet) and ResNet-32 to ensure comparability. We agree that a broader evaluation is important and are happy to report results for additional tasks and architectures in the final version.
>
> **3. The paper mentions that $\rho_k$ values in GCA losses can be tuned via cross-validation but provides limited guidance on computational complexity or sensitivity analysis. The suggestion to use $[m_k^{1/3}]_k$ as starting points without any intuitions theoretical justification.**
>
> **Response:** We followed (Cortes et al., 2025) in using $\rho_k = m_k^{1/3}$ in our experiments. A similar derivation to theirs can be adapted to our setting, showing that these values are theoretically optimal in a separable case, which provides a justification and guidance for selecting $\rho_k$ for GCA losses. Empirically, we also found GCA losses to be robust to variations in $\rho_k$ around these values. While $\rho_k$ can be tuned via cross-validation, the default choice of $m_k^{1/3}$ performs well. We will further clarify these points and include a discussion in the final version.
>
> **4. While the paper mentions training details, it doesn't thoroughly analyze the computational cost of the proposed methods compared to baselines.**
>
> **Response:** For fixed hyperparameters, the computational cost of our proposed methods is comparable to that of standard neural networks trained with cross-entropy loss (that is, logistic loss with softmax) and to that of the baselines. Our approach remains practical with commonly used optimizers such as SGD, Adam, and AdaGrad. While our methods introduce additional hyperparameters, namely $\rho_k$ and $q$ in GCA losses and $q$ in GLA losses, the value of $\rho_k$ has a default choice (as discussed above), and $q$ serves a similar role to hyperparameters in the baseline methods listed in Table 1, many of which also involve at least one extra tunable parameter. We will further elaborate on the computational cost and provide more detailed comparisons in the final version.
>
> **Questions:**
>
> **1. Line 14: "H-consistent for unbounded and complete hypothesis sets" - The conjunction "and" is confusing here. Should clarify if both conditions are required or if it's "unbounded or complete."**
>
> **Response:** Thank you. Here, "unbounded" and "complete" refer to the same condition, and "and" was intended to mean "or." We will revise the wording to clarify this in the final version.
>
> **2. Line 92: "The margin $\rho_h(x, y)$ for a predictor $h \in H$" - The margin notation $\rho$ conflicts with the imbalance ratio $\rho$ used later. Could you clarify?**
>
> **Response:** Thank you. We will revise the notation.
>
> **3. Line 130: The LA loss definition uses $\tau$ but the text doesn't clearly explain what happens when $\tau \neq 1$ until much later.**
>
> **Response:** Thank you. We will move the comment on the case $\tau \neq 1$ directly after the loss definition for better clarity.
>
> **4. Line 177: $\rho = (\rho_1, \ldots, \rho_n)$ - This reuses the margin notation from Page 3.**
>
> **Response:** Thank you. We will revise the notation to avoid this conflict.
>
> **5. Line 191: "Note that while the $\rho_k$ values can be freely tuned" - What do you mean?**
>
> **Response:** We mean that the $\rho_k$ values can be treated as tunable hyperparameters in practice, which may further improve performance. In our experiments, we use the theoretically guided values $\rho_k = m_k^{1/3}$.
>
> **6. Line 258: The bound $\Gamma(t) = \sqrt{\frac{2 t}{p_{\min}}}$  - Is this a tight bound?**
>
> **Response:** The bound can be tightened by using the exact expression, such as the inverse of the function $T$ in Theorem 3.1 of (Mao et al., 2023b), instead of the square root approximation. We used the square root form for simplicity of presentation, following the same approximation approach as in (Mao et al., 2023b).
>
> **7. Line 283: The theorem statement is quite dense and could benefit from more intuitive explanation.**
>
> **Response:** Thank you. We will use the additional page in the final version to provide more intuitive explanations of the theorems.
>
> **8. Line 343: "imbalance ratio, $\rho = \frac{\max_{k = 1}^c m_k}{\min_{k = 1}^c m_k}$" - The notation $c$ for number of classes conflicts with the usage of $n$ elsewhere.**
>
> **Response:** Thank you. We will revise the notation to consistently use $n$ for the number of classes.
>
> **9. Line 351: "200 epochs" - No justification for this choice or sensitivity analysis.**
>
> **Response:** We followed (Cortes et al., 2025) in using 200 epochs, which is a standard choice in the literature.
>
> **10. Eq (5): The confidence margin parameters $\rho_y$ should be clearly distinguished from the margin function $\rho_h(x, y)$ defined earlier.**
>
> **Response:** Thank you. We will revise the notation to clearly distinguish the two.
>
> **11. How sensitive are the GCA losses to the choice of confidence margin parameters $\rho_y$?**
>
> **Response:** We followed (Cortes et al., 2025) in using $\rho_k = m_k^{1/3}$, which can be shown to be theoretically optimal in our setting as well. Empirically, GCA losses are robust to variations around these guided values.
>
> **12. What is the computational complexity comparison between GLA and GCA losses?**
>
> **Response:**  For fixed hyperparameters, the computational cost of GLA and GCA losses is comparable to standard training with cross-entropy loss. Both involve a hyperparameter $q$, and GCA additionally uses per-class margins $\rho_k$, for which we adopt a default choice $\rho_k = m_k^{1/3}$. We will include more detailed comparisons in the final version.
>
> **13. Are there convergence guarantees for the optimization of these loss functions?**
>
> **Response:** Our loss functions are adapted from the general cross-entropy family and share similar convergence behavior when optimized with standard methods such as SGD, Adam, and AdaGrad. We will clarify this and provide more discussion in the final version.
>
> **14. It would be great if the authors clarify the proof novelty compared to Mohri et al., 2018 and Mao et al., 2023b,a.**
>
> **Response:** We refer to our response to “Weakness 1” for clarification on the proof novelty.

---

### Comment · Area_Chair_JhgQ · 2025-08-05
**Author-Reviewer Discussion Phase**

Dear Authors and Reviewers,

Thank you for supporting NeurIPS 2025!

We are now in the final two days of the discussion phase. Please make the most of this time to exchange your views!

I encourage each Reviewer to read the Authors' responses and provide a reply, if you have not already done so.

Thank you,

NeurIPS 2025 Area Chair

---

### Decision · Program_Chairs · 2025-09-17

**Decision:**

Accept (poster)

**Comment:**

The paper addresses the problem of imbalanced multi-class classification. The Authors propose two surrogate loss families: Generalized Logit-Adjusted (GLA) losses and Generalized Class-Aware weighted (GCA) losses. They analyze their Bayes- and
${H}$-consistency, showing that GLA losses are Bayes-consistent but only ${H}$-consistent for complete hypothesis sets, with bounds scaling as $1/p_{\min}$. In contrast, GCA losses are  ${H}$-consistent for both bounded and complete hypothesis sets, with more favorable bounds scaling as $1/\sqrt{p_{\min}}$.

This is a very good and interesting paper that addresses an important theoretical problem. All Reviewers are positive about the contribution. The main critical remarks concern the readability of the paper, as it is rather dense from a theoretical perspective, and the relation to previous theoretical works on similar topics. The Authors have promised to improve the writing, and they have clarified that the relation to prior work lies in extending it to a new and specific problem, which required the development of new proof techniques.